# LocDiffusion: Identifying Locations on Earth by Diffusing in the Hilbert Space

## Abstract

*Image geolocalization* is a fundamental yet challenging task, aiming at inferring the geolocation on Earth where an image is taken. Existing methods approach it either via grid-based classification or via image retrieval. The geolocalization accuracy of these methods is constrained by the choice of geographic grid cell sizes or the spatial distributions of the retrieval image/geolocation gallery, and their performance significantly suffers when the spatial distribution of test images does not align with such choices. To address these limitations, we propose to leverage diffusion models to achieve image geolocalization with arbitrary resolutions. To avoid the problematic manifold reprojection step in diffusion, we developed a novel *spherical positional encoding-decoding* framework, which encodes points on a spherical surface (e.g., geolocations on Earth) into a Hilbert space of Spherical Harmonics coefficients and decodes points (geolocations) by mode-seeking. We call this type of position encoding **Spherical Harmonics Dirac Delta (SHDD) Representation**. We also propose a novel SirenNet-based architecture called **CS-UNet** to learn the conditional backward process in the latent SHDD space by minimizing a latent KL-divergence loss. We train a conditional latent diffusion model called **LocDiffusion** that generates geolocations under the guidance of images – to the best of our knowledge, the first generative model to address the image geolocalization problem. We evaluate our LocDiffusion model against SOTA image geolocalization baselines. LocDiffusion achieves competitive geolocalization performance and demonstrates significantly stronger generalizability to unseen geolocations.

## 1 Introduction

Predicting locations on Earth based on a given condition (e.g., input image or text) is a fundamental yet challenging task. Image geolocalization, being a prominent example of this task, aims at predicting the geolocations only based on images, such as wildlife photos, street views, and remote sensing images. However, unlike image classification, solutions to image geolocalization are less mature because its ground-truths are locations represented by **real-valued** coordinates on the **spherical** surface. While regression models are commonly used to predict real-valued labels, they are proved to be tricky to train and perform especially poorly on image geolocalization due to the highly complex and non-linear mapping between the image space and the geospatial space (Vo et al., 2017; Izbicki et al., 2020). As an alternative solution, researchers employ pre-defined geographical classes (e.g. divide Earth into disjoint or hierarchical grid cells) or geo-tagged image galleries (e.g. a set of reference geotagged images) to map the real-valued ground-truth coordinates to discrete labels (e.g. the ID of the grid cell the ground-truth falls into or the ID of the reference image in the gallery that has the closest geotag as the label of a ground-truth location), subsequently transforming image geolocalization problem into a special case of image classification or image-image/image-location retrieval task. For example, both Vo et al. (2017) and CPlaNet (Seo et al., 2018) partition the Earth's surface into non-overlapping grid cells and convert the image geolocalization problem into an image classification problem. GeoCLIP (Vivanco et al., 2023) uses a contrastive learning framework to align pretrained image embeddings with geographical location embeddings in the gallery and achieves SOTA performance. However, **the spatial resolution of these approaches is constrained by the size of the grid cells or the spatial distribution of gallery images/locations.**

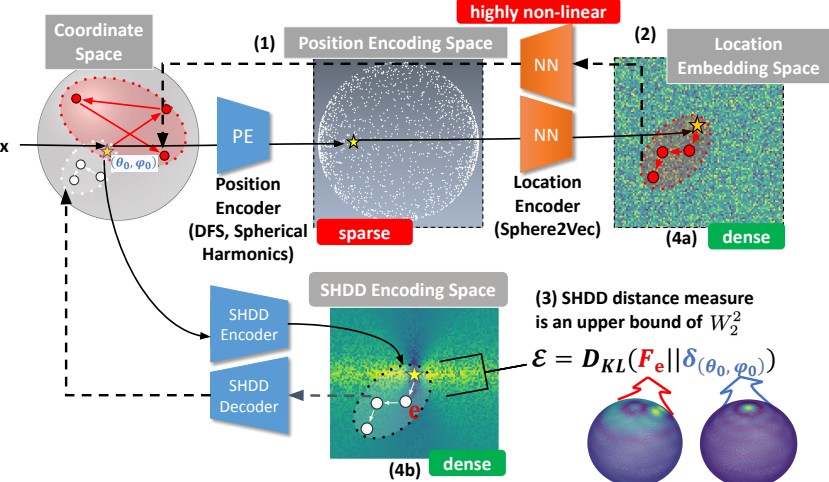

Figure 1: The difficulties of latent diffusion for image geolocalization. **Black solid/dotted arrows** denote the encoding/decoding steps. **Orange modules** are learnable, while **blue modules** are deterministic with no learning parameters. **(1)** It is difficult to diffuse in the **position encoding space** because valid positional encodings are also sparse. The diffusion model cannot function directly in the position encoding space, and learning a generalizable decoder on sparse data is also difficult. **(2)** The **locational embedding space** is dense and can perform diffusion processes, but the non-linear mapping between the position encoding and location embedding space makes decoding back to a correct coordinates extremely difficult. Minimizing distances in the location embedding space may not minimize geographic distance, and vice versa. **(3)** The SHDD encoding space is dense. Every point $\mathbf{e}$ in this encoding space corresponds to a spherical function $F_\mathbf{e}$, whose difference from the spherical Dirac delta function $\delta_{(\theta_0, \phi_0)}$ of the ground truth location $(\theta_0, \phi_0)$ is measured by the reverse KL-divergence $\mathcal{E}$. The latent diffusion in the SHDD encoding space equals gradually adding noise to $\delta_{(\theta_0,\phi_0)}$ (forward process) and find a sequence of $F_\mathbf{e}$ that gradually reduce $\mathcal{E}$ (backward process). **(4)** The SHDD decoding addresses the non-linearity problem. The heatmaps visualize the mappings from the Sphere2Vec (Mai et al., 2023b) location embedding space **(4a)** and from our SHDD encoding space **(4b)** back to the spherical coordinate space. Each pixel represents a Sphere2Vec embedding/SHDD encoding. The color of a pixel represents the distance from the spherical point represented by the embedding/encoding to the yellow star point in the middle. The mapping from the SHDD encoding space is significantly smoother.

Diffusion models have demonstrated great potential in directly and stably generating continuous outputs such as images and modeling complex distributions. They are commonly applied to points in Euclidean spaces (Song et al., 2020; Ho et al., 2020b; Song et al., 2021) or the geometric structures defined in Euclidean spaces (Xu et al., 2023). This motivates us to develop diffusion-based image geolocalization methods that output location predictions on the spherical surface with arbitrary spatial resolution and without dependence on predefined grid cells or galleries. However, naively performing *diffusion in the coordinate space* faces two major drawbacks. First, geographical locations do not form a Euclidean space. They reside on an embedded Riemannian manifold[1]. *Diffusion in the geographical coordinate space is ineffective because of projection distortion and sparsity*, i.e., performing diffusion on the XYZ coordinates will likely lead to a point that is not on the spherical surface. It is possible to perform diffusion on the manifolds, but it is very computationally expensive (Huang et al., 2022). Second, more importantly, raw coordinates cannot represent rich multi-scale geographical information or modeling complex spatial distributions (Mai et al., 2023b; Rußwurm et al., 2024). In order to achieve good modeling power for complex distributions over space, location representation methods (Mac Aodha et al., 2019; Mai et al., 2020b; 2022; 2023b; Rußwurm et al., 2024; Wu et al., 2024) commonly adopt multi-scale position encoding with deterministic transformations followed by learnable location embedding layers. [2] *Diffusion in the coordinate space would require non-standard diffusion model with multi-scale representations internally.*

---

[1]Geographical locations are distributed on a 2-dimensional Riemannian manifold (i.e., the sphere surface) embedded in the 3-dimensional Euclidean space.

[2]They share a common framework: First, they encode the point $\mathbf{p}$ into multi-scale features such as sinusoidal features (Vaswani, 2017; Mai et al., 2020b) and Double Fourier Sphere (DFS) features (Orszag, 1974; Mai et al., 2023b). Then, these models train a neural network to embed the features into dense representations via supervised learning, unsupervised learning, or contrastive learning (Mai et al., 2023a; Klemmer et al., 2023; Vivanco et al., 2023). The former step is called *position encoding* ($\mathbb{PE}$), and conventionally we call the encoded features $\mathbb{PE}(\mathbf{p})$ the *positional encoding* of $\mathbf{p}$. Similarly, the latter step is called *location embedding* ($\mathbb{NN}$) and the learned representation $\mathbb{NN}(\mathbb{PE}(\mathbf{p}))$ is called the *locational embedding* of $\mathbf{p}$.

Despite their wide applicability, neither the position encoding space nor the location embedding space is suitable for developing a location diffusion model due to **sparsity** problem during diffusion and **non-linearity** problem during decoding as illustrated in Figure 1(1)(2). On the one hand, *the position encoding space has a "sparsity problem"*. The position encoding layer commonly increase the dimensionality of the representation significantly from the coordinate space. Therefore, all valid positional encodings form a very low-dimensional manifold embedded in a high-dimensional Euclidean space. If we diffuse in the embedded high-dimensional space and train a decoder to map Euclidean points back to geographic coordinates, the sparsity of position encodings (from the available training data) makes it very difficult to learn smooth local interpolations that generalizes to unseen data. On the other hand, while the location embedding space is dense and suitable for forward and backward diffusion processes, it leads to the *difficulty to learn an inverse mapping which decodes the location embedding directly back to the coordinate space, skipping the position encoding space*, because of the non-linear mapping between position encodings and location embeddings.

We hypothesize that the ideal space to develop latent diffusion models for spherical location generation should be both dense and easy to find projections back to the coordinate space. Motivated by this observation, we propose a novel spherical position encoding method called *Spherical Harmonics Dirac Delta (SHDD) Representation*. Figure 1(3)(4) illustrates how our method addresses the sparsity problem by encoding a spherical point $(\theta_0, \phi_0)$ as a spherical Dirac delta function $\delta_{(\theta_0, \phi_0)}$. In the SHDD encoding space, every point $\mathbf{e}$ uniquely corresponds to a spherical function $F_{\mathbf{e}}$ and can be seen as a noised spherical Dirac delta function. The level of noise $\mathcal{E}$ can be continuously measured by the reverse KL-divergence between $F_{\mathbf{e}}$ and $\delta_{(\theta_0, \phi_0)}$. Then the latent diffusion in the SHDD encoding space equals gradually adding noise to the ground-truth $\delta_{(\theta_0, \phi_0)}$ (forward process) and find a sequence of $F_{\mathbf{e}}$ that gradually reduce $\mathcal{E}$ (backward process). During decoding, the learning-free SHDD Decoder evaluates the corresponding spherical function $F_{\mathbf{e}}$ and decodes it as the spherical point whose corresponding spherical Dirac delta function minimizes $\mathcal{E}$. Figure 1 4(b) demonstrates that our SHDD encoding space shows less decoding non-linearity than existing location representation learning methods such as Sphere2Vec (Mai et al., 2023b) and Rußwurm et al. (2024). Therefore, diffusion in the SHDD encoding space will be more stable and easier to converge.

Equipped with the Hilbert (i.e. infinite dimensional Euclidean) SHDD encoding space and the SHDD decoder, we can now perform conventional latent diffusion for location generation. We propose a novel SirenNet-based architecture called *Conditional Siren-UNet (CS-UNet)* to learn the conditional backward diffusion process, i.e, to generate spherical points from random Gaussian noise given conditions such as images and texts. We call the integrated framework, including SHDD encoding, CS-UNet latent diffusion, and SHDD decoding, which enables efficient conditional generation of spherical points, the *LocDiffusion* model. On global image geolocalization tasks, the performance of LocDiffusion competes with state-of-the-art models, and is proven to be more spatially generalizable than existing retrieval based geolocalization models by ablation experiments.

## 2 RELATED WORK

**Geolocalization by classification and retrieval.** Traditional geolocalization methods typically employ either a classification approach or an image retrieval approach. The former divides the Earth's surface into non-overlapping or hierarchical grid cells and classifies images accordingly (Pramanick et al., 2022a; Vo et al., 2017; Muller-Budack et al., 2018) while the later approach identifies the location of a given image by matching it with a database of image-location pairs (Shi et al., 2020; Zhu et al., 2023; Zhou et al., 2024). Using fewer cells results in lower location prediction accuracy, while using smaller grids reduces the number of training examples per class and risks overfitting (Seo et al., 2018). On the other hand, retrieval-based systems usually suffer from poor search quality and inadequate coverage of the global geographic landscape.

**Diffusion in the coordinate space.** Conventional diffusion models cannot function well in the spherical coordinate space (e.g., 3D coordinates representing points on a sphere) because valid points for diffusion are too sparse, i.e., adding or removing noise to a point on the manifold almost always results in a point outside the manifold. While certain coordinates such as latitude and longitude can remain in the valid manifold with noises, these spaces are non-Euclidean and not suitable for existing denoising diffusion implicit model (DDIM) models (Song et al., 2021). They can also cause significant distortions in localization (e.g., in regions with high latitudes).

**Riemannian diffusion models.** There are two common strategies in the past to address the above problem. The first strategy is to project a point on the sphere to its tangent space (which is Euclidean), add/remove noise in the tangent space, and re-project the noised/denoised point in the tangent space back to the surface (Rozen et al., 2021). The second strategy is to derive formulas for direct Riemannian diffusion (Huang et al., 2022). The main drawback of both strategies is their computational complexity. In the first case, each projection operation takes time, making acceleration based on DDIM (Song et al., 2021) impossible, because the projections are accurate only when the diffusion steps are adequately small. In the second case, the Riemannian diffusion formulation is much more complicated than the Euclidean version. The model architectures, training tricks, and other useful techniques developed for conventional diffusion models can not be easily transferred.

**Location Embedding.** The distinction between positional encoding and location embedding lies in semantics: the positional encoding is only a task-agnostic transformation of the coordinates of $\mathbf{x}$, but the location embedding carries task-specific information. For example, it can contain information about spatial distributions of species if trained on geo-aware species fine-grained recognition tasks (Mac Aodha et al., 2019; Mai et al., 2023b;a; Cole et al., 2023). Some prior work on location encoding, such as NeRF (Mildenhall et al., 2020), utilized positional encoding to represent location information. This task-agnostic method focuses on capturing the position or order of elements within a sequence. In contrast, many location encoders are specifically designed to capture context-aware or spatially-aware location information. These encoders can be categorized into two groups: 2D location encoders (Berg et al., 2014; Tang et al., 2015; Mac Aodha et al., 2019; Mai et al., 2020b), which operate in projected 2D space, and the other is 3D location encoders (Mai et al., 2023b; Rußwurm et al., 2024) which interpret geolocation as 3D coordinates on earth surface. Please refer to A.1 in the Appendix for more detailed information on location encoders.

## 3 PRELIMINARIES

### 3.1 REAL BASIS OF SPHERICAL HARMONICS

Let $\mathbf{p} = (\theta, \phi)$ be a location on the spherical surface using conventional angular coordinates where $\theta \in [0, \pi)$ and $\phi \in [0, 2\pi)$. For any function $F(\theta, \phi)$ on the sphere, there exists a **unique** infinite-dimensional real-valued vector of coefficients $\{C_{lm}\}$ (we may call it *coefficient vector*) such that

$$\forall (\theta, \phi), F(\theta, \phi) = \sum_{l=0}^{\infty} \sum_{m=-l}^{l} C_{lm} Y_{lm}(\theta, \phi) \tag{1}$$

where $l$ is called *degree* and $m$ is called *order* and $Y_{lm}(\theta, \phi)$ is the *real basis of spherical harmonics* at degree $l$ and order $m$. The detailed computation of $Y_{lm}$ can be found in Appendix A.2. In this way, any function on the sphere can be uniquely represented by its coefficient vector.

### 3.2 SPHERICAL DIRAC DELTA FUNCTION

Conventionally, a Dirac delta function $\delta$ is defined as a distribution on the real line where all probability mass concentrates on one single value, i.e., a single-point distribution. Analogously, a spherical Dirac delta function is a probability density function over the spherical surface whose mass all concentrates on one point:

$$\delta_{(\theta_0, \phi_0)}(\theta, \phi) = \begin{cases} \infty & \theta = \theta_0, \phi = \phi_0 \\ 0 & \text{otherwise} \end{cases} \tag{2}$$

Therefore, we can use a spherical Dirac delta function to uniquely represent any point $(\theta_i, \phi_i)$ on the sphere by mapping it to $\delta_{(\theta_i, \phi_i)}$. Representing a point as a function allows us to use spherical harmonics to represent points on the spherical surface.

## 4 LOCDIFFUSION FRAMEWORK

In this section, we will introduce the theory and techniques we employ in our LocDiffusion model that enable spherical location generation via latent diffusion. Our aim is to find a position encoding space that does not suffer from the sparsity problem and the non-linearity problem so we can

efficiently perform latent diffusion. We first analyze what properties we need to achieve this and propose the Spherical Harmonics Dirac Delta (SHDD) Encoding-Decoding framework accordingly. Then we prove that SHDD satisfies all the desired properties. Following that, we propose the Conditional Siren-UNet (CS-UNet) architecture to learn the conditional backward process for latent diffusion. We also develop computational techniques based on the properties of SHDD representation so that the training and inference of LocDiffusion are efficient.

## 4.1 PROBLEM SETUP AND INTUITIONS

As we have outlined in the introduction, our goal is to find a position encoding method that encodes the spherical surface into a dense subset of $\mathbb{R}^d$ (ideally the entire $\mathbb{R}^d$) and accurately decode points back to spherical coordinates. There are several mathematical properties such position encoding and decoding method should have. For rigorous discussions, we give definitions of the aforementioned properties and demonstrate how they guide the finding of our SHDD encoding-decoding framework.

**Definition 4.1 (Coordinate Space)** *A Coordinate Space $\mathcal{C}$ can be any space with a parametrization, such as Euclidean space with the Descartes coordinate system. In this paper, $\mathcal{C}$ always refers to the unit sphere surface embedded in $\mathbb{R}^3$ with the conventional angular coordinate system $(\theta, \phi)$.*

**Definition 4.2 (Position Encoding and Position Decoding)** *A Position Encoder $\mathbb{PE} : \mathcal{C} \to \mathbb{R}^d$ is an injective function, usually $d \gg 3$. $\mathcal{S}_{\mathbb{PE}} := \mathbb{PE}(\mathcal{C}) \subset \mathbb{R}^d$ is called the Position Encoding Space. A Position Decoder $\mathbb{PD} : \mathbb{R}^d \to \mathcal{C}$ is a surjective function.*

**The sparsity problem:** Since we are projecting a set of 2-dimensional points in $\mathcal{C}$ into a high-dimensional Euclidean space $\mathcal{S}_{\mathbb{PE}}$, dense filling is impossible. However, if we define a **difference measure** $\mathcal{E} : \mathbb{R}^d \times \mathbb{R}^d \to \mathbb{R}$, then $\mathcal{S}_{\mathbb{PE}}$ can be *partitioned* by the following equivalence relation:

$$\mathbf{e} \overset{\mathcal{E}}{\sim} \mathbf{e}' \Leftarrow \arg\min_{\mathbf{s} \in \mathcal{S}_{\mathbb{PE}}} \mathcal{E}(\mathbf{e}, \mathbf{s}) = \arg\min_{\mathbf{s} \in \mathcal{S}_{\mathbb{PE}}} \mathcal{E}(\mathbf{e}', \mathbf{s}) \tag{3}$$

that is, we can assign every point $\mathbf{s} \in \mathcal{S}_{\mathbb{PE}}$ to the nearest positional encoding (consequently, a spherical point) in terms of $\mathcal{E}$. We say the $\mathcal{E}$-equivalence classes densely fill $\mathbb{R}^d$. Further, a **learning-free decoder** exists as

$$\mathbb{PD}_{\mathcal{E}}(\mathbf{e}) := \{\mathbf{p} \in \mathcal{C} | \mathbf{e} \overset{\mathcal{E}}{\sim} \mathbb{PE}(\mathbf{p})\} = \arg\min_{\mathbf{p} \in \mathcal{C}} \mathcal{E}(\mathbf{e}, \mathbb{PE}(\mathbf{p})) \tag{4}$$

If $\mathcal{E}$ is **continuous**, i.e.

$$\forall \mathbf{s} \in \mathcal{S}_{\mathbb{PE}}, (\mathbf{e} \to \mathbf{s}) \Rightarrow (\mathcal{E}(\mathbf{e}, \mathbf{s}) \to 0) \tag{5}$$

then the sparsity problem is resolved, since now diffusion in $\mathcal{S}_{\mathbb{PE}}$ equals a random walk among spherical points and small perturbation will not result in an abrupt jump on the spherical surface.

**The non-linearity problem:** Since the diffusion model has intrinsic randomness, it is possible that the generated $\mathbf{e}$ corresponds to a wrong $\mathbf{s}$. If the mapping between $\mathbf{s}$ and its corresponding spherical point $\mathbf{p} = \mathbb{PD}_{\mathcal{E}}(\mathbf{s})$ is highly non-linear (e.g., in the location embedding space), the decoder $\mathbb{PD}_{\mathcal{E}}$ will then be very unstable (see Figure 1). Thus, we hope that for a large tolerance $\eta > 0$ and a small shift $\Delta > 0$, the following property holds for our decoder $\mathbb{PD}_{\mathcal{E}}$:

$$\forall \mathbf{s} \in \mathcal{S}_{\mathbb{PE}}, \mathcal{E}(\mathbf{e}, \mathbf{s}) < \eta \Rightarrow d_{\mathcal{C}}(\mathbb{PD}_{\mathcal{E}}(\mathbf{e}), \mathbb{PD}_{\mathcal{E}}(\mathbf{s})) < \Delta \tag{6}$$

where $d_{\mathcal{C}}$ is the distance in the spherical coordinate space (e.g., the great circle distance). If this property is satisfied, the non-linearity problem is resolved.

It is not an easy task to find such $\mathcal{E}$, especially considering computational constraints (e.g., it is impossible to exactly evaluate the $\arg\min$ function in Equation 3). Fortunately, we find that by treating spherical points as special spherical functions and represent them using Spherical Harmonics coefficients, we can define $\mathcal{E}$ as spherical KL-divergence which satisfies all the desirable properties mentioned above, thus addressing the sparsity and the non-linearity problems as a whole. Moreover, the choice of Spherical Harmonics coefficients also enables efficient computation.

## 4.2 SPHERICAL HARMONICS DIRAC DELTA (SHDD) ENCODING

As discussed in Section 3.2, we can represent spherical points as spherical Dirac delta functions. Consider Section 3.1, a spherical Dirac delta function can be encoded as an infinite-dimensional real-valued coefficient vector, i.e. a point in a Hilbert space. Here, $\bigcup$ denotes vector concatenation.

$$\mathbb{PE}_{\text{SHDD}}(\theta_0, \phi_0) := \bigcup_{l=0}^{\infty} \bigcup_{m=-l}^{l} [C_{lm}] \tag{7}$$

Thus, the spherical harmonics coefficient vector can be used to uniquely represent a point $(\theta_0, \phi_0)$ on the sphere. In practice, it is impossible to deal with infinite-dimensional coefficient vectors. It is also impossible to deal with the infinite probability density of spherical Dirac delta functions. These two practical constraints, fortunately, can be settled as a whole: we truncate the coefficient vector up to its leading $(L+1)^2$ dimensions, where $L$ is the maximum degree of associate Legendre polynomials. Therefore, the $L$-degree representation of point $(\theta_0, \phi_0)$ is defined as

$$\mathbb{PE}_{\text{SHDD}}^{L}(\theta_0, \phi_0) := \bigcup_{l=0}^{L} \bigcup_{m=-l}^{l} [C_{lm}] \tag{8}$$

We call this $(L+1)^2$-dimensional real-valued vector the $L$-degree *Spherical Harmonics Dirac Delta (SHDD) Representation* of $(\theta_0, \phi_0)$ and $\mathbb{PE}_{\text{SHDD}}$ the SHDD encoder. Each SHDD representation corresponds to an approximation of the true spherical Dirac delta function $\delta_{(\theta_0, \phi_0)}$, whose probability density concentrates in a region surrounding $(\theta_0, \phi_0)$ rather than a single point, solving the infinite density problem. The Legendre polynomials have finer granularity as their degree $L$ increases, which makes SHDD representations, like other frequency-based location encoding methods such as Sphere2Vec (Mai et al., 2023b), capable of capturing multi-scale spatial information.

Problems remain on how to find the values of $C_{lm}$. For arbitrary spherical functions, $C_{lm}$ needs to be iteratively computed. However, for spherical Dirac delta functions, we can efficiently obtain $C_{lm}$ thanks to the fact that the coefficients of the Legendre polynomials are the values of the Legendre polynomials at $(\theta_0, \phi_0)$ (Arfken et al., 2011), i.e.,

$$F \equiv \delta_{(\theta_0, \phi_0)} \Leftrightarrow \forall (\theta, \phi), F(\theta, \phi) = \sum_{l=0}^{\infty} \sum_{m=-l}^{l} Y_{lm}(\theta_0, \phi_0) Y_{lm}(\theta, \phi) \tag{9}$$

That is, for spherical Dirac delta functions, $C_{lm} \equiv Y_{lm}(\theta_0, \phi_0)$ for any $l$ and $m$. So instead of iteratively computing $\{C_{lm}\}$ in the general case, the encoding procedure can be reduced to a simple look-up of $Y_{lm}$ values.

It is worth noting that while the SHDD representation in Equation 8 has the same expression as the SH positional encoding used in a recent work (Rußwurm et al., 2024), they refer to **distinct** mathematical objects. The SH positional encoding of a point $(\theta_0, \phi_0)$ in their work is the sequence of evaluated $Y_{lm}(\theta_0, \phi_0)$ values, which forms a sparse **feature space**. The SHDD representation of a point $(\theta_0, \phi_0)$ in our work is the $C_{lm}$ values of the corresponding spherical Dirac delta function $\delta_{(\theta_0, \phi_0)}$, whose $\mathcal{E}$-equivalence classes form a Hilbert **coefficient space**. The reason that the two types of positional encodings coincidentally have identical expressions is only because spherical Dirac functions satisfy Equation 9, i.e., $C_{lm} \equiv Y_{lm}(\theta_0, \phi_0)$.

## 4.3 THE SHDD DISTANCE MEASURE

**SHDD KL-Divergence** An $L$-degree SHDD representation corresponds to a spherical Dirac delta function $\delta$, and an arbitrary $\mathbb{R}^{(L+1)^2}$ vector corresponds to certain spherical function $F$. Thus, we can use the reverse KL-divergence between (the normalized) $F$ and $\delta$ as the difference measure $\mathcal{E}$. Let $p_{(\theta, \phi)}$ and $q_{\mathbf{e}}$ be the normalized probability distributions corresponding to the SHDD representation of $(\theta, \phi)$ and an arbitrary $\mathbb{R}^{(L+1)^2}$ vector $\mathbf{e} = \bigcup_{l=0}^{L} \bigcup_{m=-l}^{l} [e_{lm}]$. Specifically,

$$p_{(\theta, \phi)}(u, v) := \exp\left(\sum_{l=0}^{L} \sum_{m=-l}^{l} Y_{lm}(\theta, \phi) Y_{lm}(u, v)\right) / Z(\mathbb{PE}_{\text{SHDD}}(\theta, \phi)) \tag{10}$$

$$q_{\mathbf{e}}(u, v) := \exp\left(\sum_{l=0}^{L} \sum_{m=-l}^{l} e_{lm} Y_{lm}(u, v)\right) / Z(\mathbf{e}) \tag{11}$$

Here $Z(\mathbf{e}) = \int_{u'=0}^{u'=\pi} \int_{v'=0}^{v'=2\pi} \exp\left(\sum_{l=0}^{L} \sum_{m=-l}^{l} e_{lm} Y_{lm}(u', v')\right) \mathbf{d}u' \mathbf{d}v'$ is a normalization constant and the exponential ensures that probabilities are non-negative. The SHDD KL-divergence

$$\mathcal{L}_{\text{SHDD-KL}}(\mathbf{e}, \mathbb{PE}_{\text{SHDD}}((\theta, \phi))) := \int_{u=0}^{u=\pi} \int_{v=0}^{v=2\pi} q_{\mathbf{e}}(u, v) \log \frac{q_{\mathbf{e}}(u, v)}{p_{(\theta, \phi)}(u, v)} \mathbf{d}u \mathbf{d}v \tag{12}$$

It is easy to verify that the SHDD KL-divergence is a continuous difference measure. As for the property described in Equation 6, notice that by Gibbs & Su (2002), the Wasserstein-2 distance $W_2$ between $p_{(\theta, \phi)}$ and $q_{\mathbf{e}}$ is bounded by the KL-divergence in the following inequality:

$$W_2^2(p_{(\theta, \phi)}, q_{\mathbf{e}}) \leq C \mathcal{L}_{\text{SHDD-KL}}(\mathbf{e}, \mathbb{PE}_{\text{SHDD}}((\theta, \phi))) \tag{13}$$

$C$ being a finite constant. $W_2$, being the Earth Mover's Distance, quantifies the amount of probability mass transport between two distributions. Thus, when $\mathcal{L}_{\text{SHDD-KL}}(\mathbf{e}, \mathbb{PE}_{\text{SHDD}}((\theta, \phi))$ is small, the difference in probability mass distribution is also small, and consequently the largest-mass-region found by the mode-seeking SHDD decoder will also remain mostly unchanged. Figure 1 visualizes this with concrete examples (pretrained Sphere2Vec location encoder and learned neural decoder v.s. our SHDD encoder and decoder) using heatmaps.

## 4.4 SHDD DECODING

**KL-Divergence SHDD Decoder** Following Equation 4, the KL-Divergence SHDD Decoder is:

$$\mathbb{PD}_{\text{KL}}(\mathbf{e}) := \arg\min_{(\theta, \phi)} \mathcal{L}_{\text{SHDD-KL}}(\mathbf{e}, \mathbb{PE}_{\text{SHDD}}((\theta, \phi))) \tag{14}$$

It is impractical to compute $\mathbb{PD}_{\text{KL}}$ exactly. Luckily, Equation 9 makes a natural simplification possible. Notice that minimizing reverse KL-divergence leads to mode-seeking behavior (Minka et al., 2005), i.e. the $(\theta, \phi)$ that satisfies Equation 14 should fall within the region with the largest probability mass. Thus, we can decode $\mathbf{e}$ by finding the center of its probability mass concentration.

**Mode-Seeking SHDD Decoder** Let $\mathbf{e} = \bigcup_{l=0}^{L} \bigcup_{m=-l}^{l} [e_{lm}]$ be an arbitrary vector in $\mathbb{R}^{(L+1)^2}$, then the position decoder $\mathbb{PD}_{\text{mode}}$ is defined as

$$\mathbb{PD}_{\text{mode}}(\mathbf{e}; \rho) := \arg\max_{(\theta, \phi)}\left\{\int_{u=\theta-\rho}^{u=\theta+\rho} \int_{v=\phi-\rho}^{v=\phi+\rho} \exp\left(\sum_{l=0}^{L} \sum_{m=-l}^{l} e_{lm} Y_{lm}(u, v)\right) \mathbf{d}u \mathbf{d}v\right\} \tag{15}$$

where $\rho$ is a hyperparameter that controls the granularity of the evaluation. There is trade-off between decoding spatial resolution and decoding stability: when $\rho$ is large, we only know the rough range of $(\theta, \phi)$ but the result is less sensitive to local spikes, and vice versa.

One advantage of adopting the SHDD decoder is its **learning-free property**. Unlike learned neural decoders, there is no loss introduced during the decoding stage. Besides, the mapping from diffusion outputs to spherical coordinates is shown to be continuous and relatively smooth. Therefore, it is safe to train latent diffusion models only using the SHDD KL-divergence loss $\mathcal{L}_{\text{SHDD-KL}}$.

Another critical advantage is that **the spatial resolution of our SHDD decoder is arbitrary** (i.e., real-valued), and not dependent on partitions of the spherical surface or the spatial distributions of image/location galleries. This is because the SHDD representation is in effect a continuous spherical function and in theory one can evaluate it in arbitrary resolution. The only two constraints are the maximum degree of Legendre polynomials $L$ which limits the spatial resolution of the spherical function itself and the computational resources (e.g., `float32` or `float64`, evaluation granularity $\rho$), both being independent from other factors.

## 4.5 CONDITIONAL SIRENNET-BASED UNET (CS-UNET) ARCHITECTURE

Inspired by the findings of Rußwurm et al. (2024), we explored different options eventually used SirenNet (Sitzmann et al., 2020) as the backbone of our diffusion model. The theoretical motivation

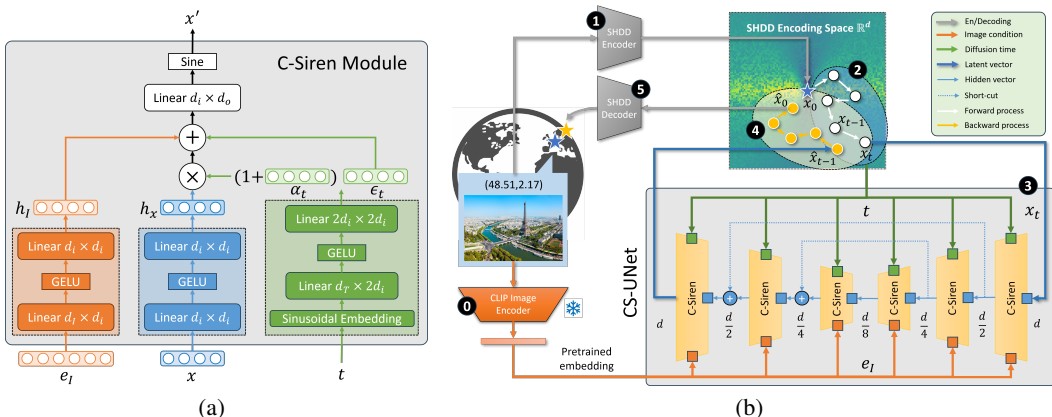

Figure 2: **(a)**: The architecture of Condition SirenNet Module (C-Siren). $x$ is the input latent vector, $x'$ is the output latent vector, $t$ is the scalar timestep, and $e_I$ is the embedding of the input image. $d_i$ is the input dimension, $d_o$ is the output dimension, $d_T$ is the time embedding dimension, $d_I$ is the conditional embedding dimension. **(b)**: The architecture of Conditional SirenNet-Based UNet (CS-UNet) and the workflow of LocDiffusion. $d$ is the latent dimension. The numbered circles denote the order of training steps.

behind this decision is that Spherical Harmonics coefficients are sums of sinusoidal and cosinusoidal functions (See Appendix A.2). Using sine as the activation function helps preserve gradients because the derivatives of sinusoidal/cosinusoidal functions are still sinusoidal and cosinusoidal functions.

Figure 2(a) depicts the network architecture of the Conditional SirenNet (C-Siren) module. The design is straightforward: inputs are the latent vector $x$, the image condition embedding $e_I$, and the diffusion step $t$. First, we use feed-forward layers to project $x$ and $e_I$ into hidden vectors $h_x$, $h_I$. Then we use the sinusoidal embedding layer (Song et al., 2021) and feed-forward layers to project the discrete diffusion timestep $t$ into a scale vector $\alpha_t$ and a shift vector $\epsilon_t$. Then, we transform $h_x$ into $h_x = (1 + \alpha_t) \odot h_x + \epsilon_t$, which is an unconditional denoising step. Following that, we sum the transformed $h_x$ and the condition $h_I$ and pass the sum to a feed-forward layer, which adjusts the denoising step under the guidance of the condition. Finally, output the sine-activated hidden vector to the next C-Siren module. Figure 2(b) completely describes the architecture of the Conditional SirenNet-Based Unet (CS-UNet).

### 4.6 LOCDIFFUSION

Next, we introduce the training cycle of our LocDiffusion model as illustrated in Figure 2(b). A training data sample includes an input image $\mathbf{I}$ and its associated geolocation $\mathbf{p} = (\theta, \phi)$ serving as the prediction target. First, we use a frozen CLIP-based image encoder (Radford et al., 2021) to encode the image $\mathbf{I}$ into an image embedding $e_I$. Then, we encode the geolocation $\mathbf{p} = (\theta, \phi)$ into its SHDD representation $\mathbb{PE}_{SHDD}(\theta, \phi)$ and store them in a look-up table. Following that, we perform a standard DDPM training (Ho et al., 2020a) based on the proposed CS-UNet architecture as shown in Figure 2(b). In a forward pass in the latent diffusion process, the spherical Dirac delta function $\delta_{(\theta_0, \phi_0)}$ defined by $\mathbb{PE}_{SHDD}(\theta, \phi)$ will be gradually added noise until being reduced to a vector whose values in each dimension are purely generated from Gaussian noise. In a backward pass of the latent diffusion model, the CS-UNet will start with a noise vector and gradually recover the $\delta_{(\theta_0, \phi_0)}$ spherical function. We implement the DDPM algorithm based on the open-source PyTorch implementation[3]. We use the SHDD KL-divergence $\mathcal{L}_{SHDD-KL}$ between the ground-truth SHDD representation $\mathbb{PE}_{SHDD}(\theta, \phi)$ and the diffusion output as the training objective, because it is more computationally stable and preserves the spatial multi-scalability than the spherical MSE (e.g. great circle distance) loss. During inferencing, we sample coefficient vectors from Gaussian noise conditioned on CLIP-based image embeddings and use $\mathbb{PD}_{mode}$ to predict locations.

There are two important implementation details worth mentioning. In practice, the integrals in Equation 12 and Equation 15 are approximated by summation. More specifically, we select a set of $N$ anchor points $\mathcal{A}_N = \{(\theta_i, \phi_i) \in \mathcal{C}\}_{i=1}^N$ on the sphere, and

$$\mathcal{L}_{SHDD-KL}(\mathbf{e}, \mathbb{PE}_{SHDD}(\theta, \phi)) = \sum_{i=1}^N q_{\mathbf{e}}(\theta_i, \phi_i) \log \frac{q_{\mathbf{e}}(\theta_i, \phi_i)}{p_{(\theta, \phi)}(\theta_i, \phi_i)} \tag{16}$$

---

[3]https://github.com/lucidrains/denoising-diffusion-pytorch

$$\mathbb{PD}_{\mathrm{mode}}(\mathbf{e};\rho) = \arg\max_{(\theta,\phi)} \sum_{i=1}^{N} \mathbb{I}\{d_{\mathcal{C}}((\theta,\phi),(\theta_i,\phi_i)) < \rho\} \exp\left(\sum_{l=0}^{L}\sum_{m=-l}^{l} e_{lm}Y_{lm}(\theta_i,\phi_i)\right) \quad (17)$$

$\mathcal{L}_{\text{SHDD-KL}}$ is used for training, thus we random sample $N = 2048$ anchor points over the globe for each mini-batch to avoid overfitting. As for $\mathbb{PD}_{\mathrm{mode}}$, the choice of $\mathcal{A}_N$ introduces inductive bias – the regions with more anchor points have heavier impact on the decoding results and higher spatial resolutions. However, Table 2 shows that LocDiffusion performs stably well on different $\mathcal{A}_N$.

## 5 EXPERIMENTS

Table 1: Main experimental results. Evaluation setup is identical to Vivanco et al. (2023). The GeoCLIP model retrieves locations from the 100k gallery provided in its code-base which aligns well with the spatial distribution of test images. $L$ is the degree of SHDD representations used in our model. **Bold** numbers denote the best performance on the corresponding dataset.

| Dataset | Model | Street 1 km | City 25 km | Region 200 km | Country 750 km | Continent 2500 km |
|---------|-------|-------------|------------|----------------|-----------------|--------------------|
| Im2GPS3k | [L]kNN, $\sigma$=4 (Vo et al., 2017) | 7.2 | 19.4 | 26.9 | 38.9 | 55.9 |
| | PlaNet (Weyand et al., 2016) | 8.5 | 24.8 | 34.3 | 48.4 | 64.6 |
| | CPlaNet (Seo et al., 2018) | 10.2 | 26.5 | 34.6 | 48.6 | 64.6 |
| | ISNs (Muller-Budack et al., 2018) | 10.5 | 28.0 | 36.6 | 49.7 | 66.0 |
| | Translocator (Pramanick et al., 2022b) | 11.8 | 31.1 | 46.7 | 58.9 | 80.1 |
| | GeoDecoder (Clark et al., 2023) | 12.8 | 33.5 | 45.9 | 61.0 | 76.1 |
| | GeoCLIP (Vivanco et al., 2023) | 14.1 | 34.5 | 50.7 | 69.7 | 83.8 |
| | PIGEON (Haas et al., 2024) | 11.3 | **36.7** | 53.8 | 72.4 | 85.3 |
| | **Ours** ($L$=47) | 10.9 | 34.0 | 53.3 | 72.5 | 85.2 |
| | **Ours** ($L$=47) + GeoCLIP | **14.4** | 35.8 | **56.4** | **73.3** | **85.5** |
| YFCC-26k | PlaNet (Weyand et al., 2016) | 4.4 | 11.0 | 16.9 | 28.5 | 47.7 |
| | ISNs (Muller-Budack et al., 2018) | 5.3 | 12.3 | 19.0 | 31.9 | 50.7 |
| | Translocator (Pramanick et al., 2022b) | 7.2 | 17.8 | 28.0 | 41.3 | 60.6 |
| | GeoDecoder (Clark et al., 2023) | 10.1 | 23.9 | 34.1 | 49.6 | 69.0 |
| | GeoCLIP (Vivanco et al., 2023) | 11.6 | 22.2 | 36.7 | 57.5 | 76.0 |
| | PIGEON (Haas et al., 2024) | 10.5 | **25.8** | **42.7** | **63.2** | **79.0** |
| | **Ours** ($L$=47) | 9.6 | 22.8 | 37.5 | 58.6 | 76.8 |
| | **Ours** ($L$=47) + GeoCLIP | **11.9** | 23.4 | 39.0 | 58.9 | 77.3 |

Table 2: Generalizability experiment results on Im2GPS3k Dataset. **Bold** numbers denote the best results obtained in the given model and gallery/anchor setting. Numbers in the brackets denote the percentage performance degrdation relative to the prior knowledge gallery/anchor.

| Model | Gallery/Anchor | Size | Street 1 km | City 25 km | Region 200 km | Country 750 km | Continent 2500 km |
|-------|----------------|------|-------------|------------|----------------|-----------------|--------------------|
| GeoCLIP | MP16 | 100 k | **14.11** | **34.47** | **50.65** | **69.67** | **83.82** |
| | Grid | 1 M | 0.03 (↓99.79%) | 9.18 (↓73.37%) | 33.47 (↓33.90%) | 55.32 (↓20.63%) | 75.34 (↓10.11%) |
| | | 500 k | 0.03 (↓99.79%) | 7.17 (↓79.21%) | 29.40 (↓41.96%) | 52.29 (↓24.94%) | 73.11 (↓12.80%) |
| | | 100 k | 0.00 (↓100.00%) | 2.67 (↓92.25%) | 22.39 (↓55.81%) | 47.35 (↓32.05%) | 68.77 (↓17.94%) |
| | | 21 k | 0.00 (↓100.00%) | 0.87 (↓97.48%) | 19.55 (↓61.41%) | 43.78 (↓37.17%) | 64.33 (↓23.26%) |
| **Ours** ($L$=23) | MP16 | 100 k | 0.57 | 11.1 | 44.42 | 68.35 | 82.50 |
| | Grid | 1 M | 0.01 (↓98.25%) | 4.37 (↓60.63%) | 43.04 (↓3.10%) | 68.30 (↓0.07%) | 81.66 (↓1.02%) |
| | | 500 k | 0.07 (↓87.72%) | 4.47 (↓59.73%) | 43.18 (↓2.79%) | **68.36** (↑0.01%) | 81.65 (↓1.03%) |
| | | 100 k | 0.07 (↓87.72%) | 4.04 (↓63.60%) | 42.91 (↓3.40%) | 68.34 (↓0.01%) | 82.18 (↓0.39%) |
| | | 21 k | 0.03 (↓94.74%) | 4.90 (↓55.86%) | 43.44 (↓2.21%) | 68.29 (↓0.09%) | 81.68 (↓0.99%) |

Table 3: Training Set-up

| Degree $L$ | Dimensions | | | Hyperparameters | | | | | | |
|------------|------------|------|-------|------------------|------|--------|-------|--------------|---------|-------------|
| | $d$ | $d_I$ | $d_T$ | batch size | lr | epochs | beta | weight decay | dropout | anchor size |
| 15, 23, 31 | 256, 576, 1024 | 768 | 200 | 512 | 0.0001 | 500 | [0.9,0.99] | 0.0005 | 0.3 | 2048 |

### 5.1 EXPERIMENTAL SETUP

We in general follow the experimental setup of GeoCLIP Vivanco et al. (2023), the SOTA model for image geolocalization, for a fair comparison. The training dataset is MP16 (MediaEval Placing Tasks 2016, Larson et al. (2017)) containing 4.72 million geotagged images. The test datasets are Im2GPS3k (Hays & Efros, 2008) and YFCC26k (Thomee et al., 2016). The GWS15k (Clark et al.,

2023) reported in GeoCLIP is unfortunately not publicly available. For each test image, our model conditionally generate 16 locations and use their geographical center as the prediction. Then we count how many predictions fall into the neighborhoods of the ground-truth locations at different scales (1 km, 25 km, 200 km, 750 km and 2500 km) respectively. Table 3 lists the details of our training setup. We use an Adam optimizer.

## 5.2 MAIN RESULTS

Table 1 summarizes the geolocalization performance of our LocDiffusion model against baselines. On Continent (2500 km), Country (750 km), and Region (200 km) levels, our model can outperform the SOTA GeoCLIP model. On the finer scales (1 km and 25 km), however, we show inferior performance due to the restricted spatial resolution of SHDD representation. With $L = 47$, the intrinsic variance of the SHDD decoder is around 200 km, making predictions on the 1 km and 25 km scales less reliable. We present a detailed analysis of how the choice of $L$ affects the performance in Appendix A.3. Instead of unlimitedly increase $L$ for higher spatial resolution, we find that combining the advantages of LocDiffusion and retrieval-based models such as GeoCLIP is more efficient: we use LocDiffusion to generate candidate locations, and restrict the retrieval of GeoCLIP to the 200 km radius region around the candidate locations. The performance improves on all scales compared to using solely LocDiffusion or GeoCLIP.

Beyond performance numbers, the biggest advantage of generative geolocalization over traditional classification/retrieval-based geolocalization methods is that it completely gets rid of predefined spatial classes and location galleries. As is admitted in Vivanco et al. (2023), the performance of retrieval-based geolocalization methods depends heavily on the quality of the gallery – i.e., how well the candidate locations in the gallery cover the test locations. For example, GeoCLIP uses a 100k gallery with locations drawn from MP16 training data. When using this gallery for the GWS15k dataset, the performance drops because there are unseen locations. It was also noticed that GeoCLIP's performance drops when an evenly sampled grid on Earth is used. At small scales this is explainable because the grids are too coarse to predict 1 km to 25 km objects. However, at large scales, the performance of GeoCLIP should not be significantly affected, but it is not the case. See the results in Table 2. With 1 million grid points, the average distance between two candidates is less than 30 km. However, the performance of GeoCLIP at the 200 km, 750 km and 2500 km scales (way larger than 30 km) is still much lower than the performance when using 100K MP16 gallery locations. It indicates that the decline in performance is due to GeoCLIP's weak generalization to new, unseen locations. We can see that the gallery has a strong inductive bias that narrows the spatial scope, and makes the retrieval model easier to overfit, but hurts its spatial generalizability.

Our LocDiffusion model, though also uses anchor points for decoding (training is random), is almost unaffected by the choice of anchor points. To align with GeoCLIP, we use the same MP16 gallery and evenly sample grid points as decoding anchor points. We can see, at the smaller scales, just like GeoCLIP, introducing the MP16 gallery helps improve the accuracy because its spatial inductive bias helps offset the vagueness of decoding. However, at larger scales, the performance of LocDiffusion is almost independent of the choice of anchors – both the way how we pick the anchor points (MP16 or even grid) and the total number of anchor points (from 21k to 1M). It is a strong indicator of better spatial generalizability for LocDiffusion.

## 6 CONCLUSION, LIMITATIONS AND FUTURE WORK

In this paper, we propose a novel SHDD encoding-decoding framework that enables latent diffusion for spherical location generation. We also propose a CS-UNet architecture to learn conditional diffusion and train a LocDiffusion model that addresses the image geolocalization task via generation. LocDiffusion achieves competitive geolocalization performance and demonstrates significantly better spatial generalizability. The major limitation of this work is that to accurately generate locations at finer scales, we need to quadratically increase the SHDD encoding dimension, which is computationally demanding. We aim to explore solutions such as hierarchical generation and random SHDD representations that reduce the space complexity from $L^2$ to linear.

**Ethics Statement**   All datasets we use in this work including the MP16, Im2GPS3k, and YFCC-26k datasets are publicly available datasets. No human subject study is conducted in this work. We do not find specific negative societal impacts of this work.

**Reproducibility Statement**   Our source code has been uploaded as a supplementary file to reproduce our experimental results. The implementation details of the spectral encoder are described in Section 4.5 and 4.6. The hyperparameters used for LocDiffusion are shown in Table 3.

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

## A APPENDIX

### A.1 SPARSITY OF EXISTING POSITIONAL ENCODING METHODS

Almost all location encoders can be formulated as the following equation (Wu et al., 2024):

$$Enc(\theta, \phi) = \mathbf{NN}(\mathbb{PE}(\theta, \phi)), \tag{18}$$

$\mathbb{PE}()$ is a position encoder that transforms the location $\mathbf{p} = (\theta, \phi)$ into a $W$-dimensional vector, referred to as the position embedding. The neural network $\mathbf{NN}() : \mathbb{R}^W \to \mathbb{R}^d$ is a learnable function that maps the position embedding $\mathbb{PE}(\theta, \phi) \in \mathbb{R}^W$ to the location embedding $Enc(\theta, \phi) \in \mathbb{R}^d$.

**1)** $tile$ is a vanilla location encoder used by many pioneering studies(Berg et al., 2014; Adams et al., 2015; Tang et al., 2015). It divides geographic regions into discrete global grids based on longitude and latitude and learns corresponding partition embeddings based on the grid cell indicator vectors.

**2)** $wrap$ (Mac Aodha et al., 2019) is a sinusoidal location encoder, normalizing latitude and longitude and processing with sinusoidal functions before feeding into $\mathbf{NN}^{wrap}()$, which is composed of four residual blocks implemented through linear layers.

**3)** $wrap + ffn$ (Mai et al., 2023b) is a variant of $wrap$ that substitutes $\mathbf{NN}^{wrap}()$ with $\mathbf{NN}^{ffn}()$, a simple FFN.

**4)** $rbf$ (Mai et al., 2020b) is a kernel-based location encoder. It randomly selects $W$ points from the training dataset as Radial Basis Function (RBF) anchor points. It then applies Gaussian kernels to each anchor points.Each input point $\vec{x}_i$ is represented as a $W$-dimension feature vector using these kernels, which is then processed by $\mathbf{NN}^{ffn}()$.

**5)** $rff$ stands for *Random Fourier Features* (Rahimi et al., 2007) and it is another kernel-based location encoder. It first encodes location $\vec{x}$ into a $W$ dimension vector - $\mathbb{PE}^{rff}(\vec{x}) = \varphi(\vec{x})$. Each component of $\varphi(\vec{x})$ first projects $\vec{x}$ into a random direction $\omega_i$ and makes a shift by $b_i$. Then it wraps this line onto a unit circle in $\mathbb{R}^2$ with the cosine function. $\mathbb{PE}^{rff}(\vec{x})$ is further fed into $\mathbf{NN}^{ffn}()$ to get a location embedding.

**6)** *Space2Vec-grid* and *Space2Vec-theory* (Mai et al., 2020b) are two versions of sinusoidal multi-scale location encoders on 2D Euclidean space. Both of them implement the position encoder $\mathbb{PE}(\vec{x})$ as performing a Fourier transformation on a 2D Euclidean space then fed into the $\mathbf{NN}^{ffn}()$. *Space2Vec-grid* treats $x = (\lambda, \varphi)$ as a 2D coordinate while *Space2Vec-theory* be simulated by summing three cosine grating functions oriented 60 degree apart.

**7)** $xyz$ (Mai et al., 2023b) is a vanilla 3D location encoder, converting the lat-lon spherical coordinates into 3D Cartesian coordinates centered at the sphere center with position encoder $\mathbb{PE}^{xyz}(\vec{x})$, then feeds the 3D coordinates into an MLP $\mathbf{NN}^{ffn}()$.

**8)** $NeRF$ can be viewed as a multiscale version of $xyz$ by employing Neural Radiance Fields (NeRF) (Mildenhall et al., 2020) as its position encoder.

**9)** *Sphere2Vec* (Mai et al., 2023b), including *Sphere2Vec-sphereC*, *Sphere2Vec-sphereC+*, *Sphere2Vec-sphereM*, *Sphere2Vec-sphereM+*, and *Sphere2Vec-dfs*, is a series of multi-scale location encoders for spherical surface based on Double Fourier Sphere (DFS) and *Space2Vec*. The multi-scale representation of *Sphere2Vec* is achieved by one-to-one mapping from each point $x_i = (\lambda_i, \varphi_i) \in \mathbb{S}^2$ with $S$ be the total number of scales. They are the first location encoder series that preserves the spherical surface distance between any two points to our knowledge.

**10)** *Siren (SH)* (Rußwurm et al., 2024) is a more recently proposed spherical location encoder, which claims a learned Double Fourier Sphere location encoder. It uses spherical harmonic basis functions as the position encoder $\mathbb{PE}^{Siren\,(SH)}(\vec{x})$, followed by a sinusoidal representation network (SirenNets) as the $\mathbf{NN}()$.

These existing location embedding spaces all suffer from sparsity issues, primarily due to the inherent correlations among the different dimensions of the position encoders. The dimensions of position embeddings are frequently interdependent. As a result, many points in the position embedding space become distant or isolated from one another.

## A.2 COMPUTATION OF SPHERICAL HARMONICS

To compute $Y_{lm}$, one can use the following expression in terms of *associated Legendre polynomials* $P_l^m(x)$:

$$Y_{lm}(\theta, \phi) = \begin{cases} (-1)^m \sqrt{2} \mathcal{J} P_l^{|m|}(\cos\theta) \sin(|m|\phi) & m < 0 \\ \mathcal{J} P_l^{|m|}(\cos\theta) & m = 0 \\ (-1)^m \sqrt{2} \mathcal{J} P_l^{|m|}(\cos\theta) \cos(|m|\phi) & m > 0 \end{cases} \tag{19}$$

where $\mathcal{J} = \sqrt{\dfrac{2l+1}{4\pi} \dfrac{(l-|m|)!}{(l+|m|)!}}$ and $P_l^m(x)$ is further computed by

$$P_l^m(x) = (-1)^m \cdot 2^l \cdot (1-x^2)^{m/2} \cdot \sum_{k=m}^{l} \frac{k!}{(k-m)!} x^{k-m} \binom{l}{k} \binom{(l+k-1)/2}{l} \tag{20}$$

## A.3 SPATIAL RESOLUTION OF SHDD ENCODING/DECODING

The spatial resolution of SHDD encoding/decoding (i.e., on what scales the mode-seeking decoder can accurately locate the probability mass concentration of the spherical Dirac delta functions) is bound by the degree $L$ of Legendre polynomials. For an $L$-degree SHDD representation, the spatial scale threshold at which it can accurately approximate spherical functions is $\pi/L$ in radian or approximately $20000/L$ in kilometers (Ince et al., 2019). For example, for $L = 15, 23, 31$, the thresholds are around 1300 km, 870 km, and 640 km, respectively. At scales significantly below half this threshold, even if the diffusion model generates accurate coefficient vectors, the mode-seeking decoder can still only decode vague locations with large variances. Figure 3 gives a visual intuition.

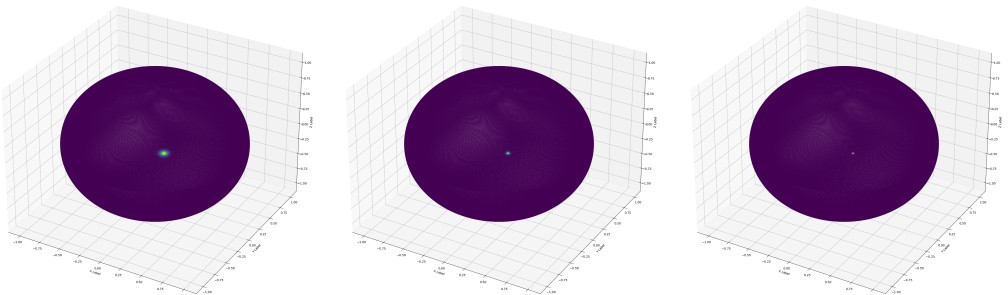

Figure 3: Illustration of the spatial resolutions with $L = 15$, $L = 23$ and $L = 31$. The bright regions are the probability mass concentrations and points within these regions are similarly likely to be decoded as the location predictions. The smaller the bright regions are, the lower errors the SHDD decoding brings.

Therefore, to uplift the performance of LocDiffusion, one straightforward way is to use larger $L$. We conduct an ablation study of the effect of $L$ on image geolocalization performance on the Im2GPS3K dataset. The results are shown in Figure 4(a). We can see as $L$ increases, while the model performances at larger spatial scales (e.g., 750km, 2500km) only increase slightly, the performances at smaller scales (e.g., 1km, 25km, 200km) see huge uplifting. This validates our hypothesis – *a larger $L$ can make the mode-seeking decoder decoding vague locations with smaller variances, thus leading to higher image geolocalization performance.* The largest $L$ we tried in Figure 4(a) is 47 which corresponds to a spatial resolution of $200km$. This is why we see huge performance improvements on the $25km$ and $200km$ curves but not on the $1km$ curve since $1km$ is still significantly smaller than the current spatial scale threshold.

However, it is not recommended to unlimitedly increase $L$. There are two major reasons:

1. The SHDD encoding dimension increases quadratically with $L$, i.e., we need quadratic space to halve the spatial resolution. It is expensive and difficult to train a diffusion model on very large encodings (e.g. to achieve 50 km spatial resolution, we theoretically need 160,000 dimensions).

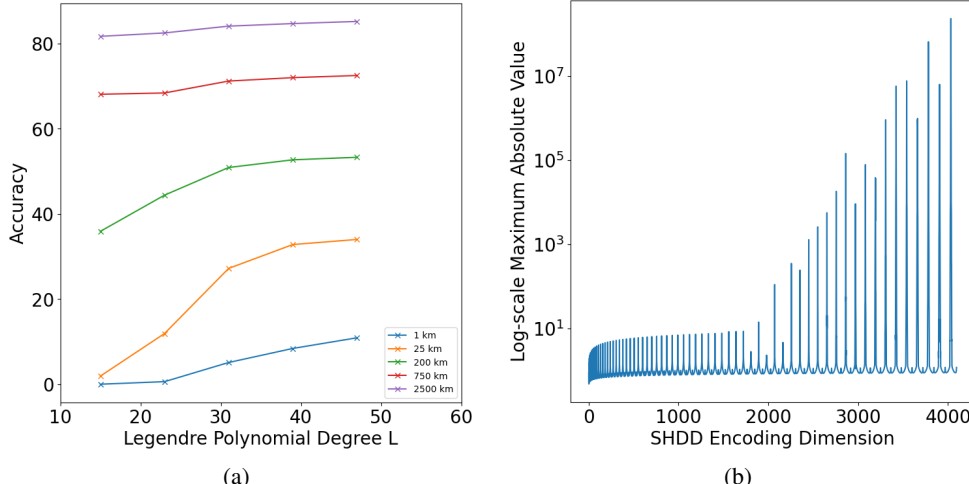

(a)            (b)

Figure 4: **(a)**: An illustration of how the image geolocalization performance on the Im2GPS3K dataset increases as L increases. Different curves indicate performance metrics on different spatial scales. **(b)**: A log-scale plot of the maximum absolute values of each SHDD encoding dimension up to $64 \times 64 = 4096$ dimensions.

2. We find that the higher the dimension of SHDD encodings, the higher the maximum absolute values of the coefficients. Figure 4(b) is a log-scale plot of the maximum absolute values of each SHDD encoding dimension up to $L = 63$ (i.e., in total $64 \times 64 = 4096$ dimensions). The absolute values below 2500 dimensions are in general manageable with only a few spikes. However, dimensions beyond this threshold become unbearably large, which makes the probability computation very unstable and easy to overflow.

Based on these observations, we use up to $L = 47$ in our paper because now the dimension of SHDD encoding goes to 2304, still within the manageable range.

Moreover, to address the high dimension issue when we use a large $L$, we find that applying a low-pass filter to the dimensions is a good dimension reduction solution. See Figure 4(b). Many dimensions of the SHDD encodings have very small absolute values and will not significantly influence the results of SHDD encoding/decoding. Thus, we can set a low-pass filter analogous to Fourier transformation and signal processing, which only keeps the dimensions that have adequately large coefficient values.

## A.4 INDUCTIVE BIAS OF GALLERY

The key factor that constrains the spatial generalizability of retrieval-based geolocalization models is the inductive bias introduced by the image gallery. When the spatial distribution of the gallery's image locations aligns well with the image locations in the test dataset, the performance of the retrieval-based models will be boosted, especially on low-error scales. However, without such inductive bias (e.g., using evenly spaced grid points as gallery locations), the performance of the retrieval-based models on all scales will suffer.

To better understand what the inductive bias of an image gallery is and how heavily it affects retrieval-based models, we calculate the statistics that demonstrate how spatially aligned the MP16 gallery used in GeoCLIP is with the Im2GPS3K test data. We measure how close test image locations are to the gallery image locations by counting the number of gallery locations that are within 1km/25km from a given test image location. Table 4 shows the statistics results. We can see that the MP16 image gallery's locations indeed closely match the image locations in the Im2GPS3K test dataset. In contrast, when we use a set of grid locations, there are much less locations falling into the 1km or 25 km buffer of the testing image locations.

Figure 5 is a set of visualizations of Table 2. It clearly demonstrates how GeoCLIP suffers greatly from using a grid gallery without prior knowledge (i.e., without using the inductive bias brought by the MP16 image gallery), while our method remains almost unaffected on larger spatial scales

Table 4: The percentage of test locations that are close (within 1 km/25 km) to multiple gallery locations.

| Gallery | MP16 | | | | Grid | | | |
|---|---|---|---|---|---|---|---|---|
| # Gallery Locations | > 1 | > 10 | > 50 | > 100 | > 1 | > 10 | > 50 | > 100 |
| Within 1 km | 63.5% | 32.7% | 14.9% | 9.78% | 0.1% | 0.0% | 0.0% | 0.0% |
| Within 25 km | 95.2% | 75.7% | 51.9% | 42.0% | 38.9% | 0.0% | 0.0% | 0.0% |

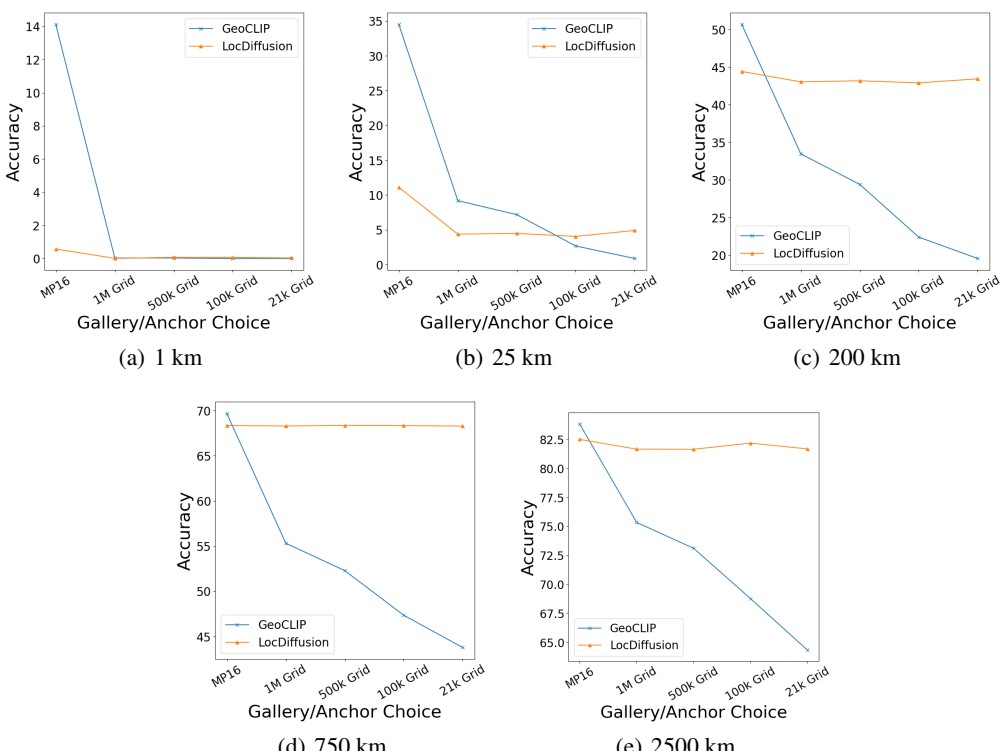

(a) 1 km     (b) 25 km     (c) 200 km

(d) 750 km     (e) 2500 km

Figure 5: From **(a)** to **(e)**: the performance changes over different choices of galleries/anchor points. Different plots indicate performance metrics under different spatial scales. For each plot, the X-axis indicates the choice of gallery locations, starting from MP16 points, 1 million grid points, to 21K grid points. The y-axis indicates the geolocalization performances on the corresponding spatial scale.

(200 km, 750 km, and 2500 km) and much less affected on smaller scales (25 km). These results clearly demonstrate that the high performance of GeoCLIP on smaller spatial scales is based on the fact that the MP16 image gallery used by GeoCLIP already contains candidate locations that are close enough to true answers (i.e., test image locations). However, this is not the case for our method because our model does not rely on such an image gallery either during training or during inferencing time. Thus, our LocDiffusion model suffers much less when we switch to a grid location gallery. Moreover, when we decrease the number of points in the grid gallery, the performances of GeoCLIP decrease significantly while the performances of our LocDiffusion are almost unaffected.

## A.5 COMPUTATIONAL COMPLEXITY

We trained our model on a Linux server equipped with four NVIDIA RTX 5500 GPUs, each with 24GB of memory. We report the training time and space complexity on a single GPU in Table 5. We do not have the training times for baseline models such as GeoCLIP and PIGEON because we did not train them from scratch and such statistics are not reported in their papers.

It can be seen that 1024 is the maximum SHDD dimension a single GPU can handle due to GPU memory constraints. For LoDiffusion models with SHDD dimensions beyond 1024, we either use

the low-pass filtering technique mentioned in Section A.3 to reduce the dimension to 1024, or split the computation across multiple GPUs. Therefore their computational complexity is not separately reported.

Table 5: Training time and space complexity. Each epoch undergoes 1500 iterations.

| Degree $L$ | Hidden Dimension | Second/Epoch | Memory (MB) |
|---|---|---|---|
| 15 | 256 | 130 | 5691 |
| 23 | 576 | 212 | 10599 |
| 31 | 1024 | 388 | 17407 |

The major factor that decides the inference time of LocDiffusion is the choice of the sampler. In our experiment, we use the original DDPM sampler (i.e., no DDIM acceleration) with 100 sampling steps. The inference time per image for LocDiffusion is 0.056s and for GeoCLIP 0.024 seconds.

## A.6 ABLATION STUDIES

### A.6.1 COMPARISON WITH OTHER LOCATION ENCODING/DECODING TECHNIQUES

As we have discussed, the superiority of using SHDD for location decoding is that its encoding space is smoother than other location encoders that use neural networks such as rbf and Sphere2Vec (Mai et al., 2023b). To demonstrate this, we evenly sample 1 million locations on Earth, encode them into corresponding location embeddings by using rbf and Sphere2Vec location encoder, and train a neural network decoder to map the location embeddings back to locations. We also use the learned neural decoder in the LocDiffusion training with weights frozen. The ablation study results are shown in Table 6. We can see that the performances of rbf and Sphere2Vec are much worse than SHDD, especially on smaller scales. This is because: (1) the learned decoder is not 100% accurate, i.e. it may decode an encoding to a wrong location, and (2) if the encoding gets a small perturbation, the decoded location may have a very large drift due to non-linearity.

Table 6: Comparing the performance of different encoders/decoders on Im2GPS3K. The **NN** is a 6-layer FFN trained on 1 million corresponding location encodings evenly spaced on Earth.

| Encoder | Decoder | 1 km | 25 km | 200 km | 750 km | 2500 km |
|---|---|---|---|---|---|---|
| rbf (Mai et al., 2020a) | NN | 0.0 | 0.0 | 18.2 | 44.1 | 60.2 |
| Sphere2Vec (Mai et al., 2023b) | NN | 0.0 | 0.0 | 22.1 | 58.4 | 72.3 |
| SHDD | SHDD | 10.9 | 34.0 | 53.3 | 72.5 | 85.2 |

To better understand the spatial drift part, Table 7 shows how much spatial drift will bring to the decoded locations when we add a small Gaussian noise (variance = 0.01) to the corresponding location encoding. We can see that compared with SHDD, both pretrained rbf and Sphere2Vec models can have much larger spatial drifts when we add a small Gaussian noise (variance = 0.01) to the corresponding location encoding. The larger the spatial drift, the less robust the encoding/decoding process is to small hidden space perturbations. Since the diffusion model will not generate perfectly noiseless encodings, such spatial drift indicates the intrinsic error of the corresponding location encoding/decoding method.

Table 7: Comparing the spatial drifts when applied a small Gaussian noise (variance = 0.01) to the encoding. The **NN** is a 6-layer FFN trained on 1 million corresponding location encodings evenly spaced on Earth.

| Encoder | Decoder | Perturbation Drift |
|---|---|---|
| rbf | NN | 102.4 km |
| Sphere2Vec | NN | 89.1 km |
| SHDD | SHDD | 5.3 km |

### A.6.2 COMPARISON WITH OTHER LOSSES

Table 8: Comparing the performance of using different training losses on Im2GPS3K.

| Loss | 1 km | 25 km | 200 km | 750 km | 2500 km |
|---|---|---|---|---|---|
| L1 | 0.0 | 0.5 | 20.3 | 30.6 | 43.5 |
| L2 | 0.0 | 0.7 | 20.1 | 32.7 | 44.9 |
| Cosine | 7.5 | 32.2 | 53.0 | 71.5 | 84.9 |
| SHDD KL-divergence | 10.9 | 34.0 | 53.3 | 72.5 | 85.2 |

Table 8 shows an ablation study on the impact of different loss functions. We can see that the SHDD KL-divergence is significantly better than L1/L2 losses. Cosine distance, being similar to our SHDD KL-divergence in terms of mathematical formulation (SHDD KL-divergence is the sum of exponential element-wise multiplications, while cosine similarity is the sum of raw element-wise multiplications), has comparable performance especially on larger scales. It would be a good approximation to reduce computational costs. We will add more thorough experiments in the camera-ready version.

### A.6.3 ABLATION STUDIES ON OTHER MODULES

We investigate how variations in the width of the CS-UNet affect its performance (see Table 9). In general, shrinking the bottleneck width $w$ of the CS-UNet seems to help alleviate model overfitting (we can adopt a lower dropout rate) and slightly boost performance, but make the model more difficult to train.

Table 9: The bottleneck width $w$ is the narrowest part of each C-Siren module. We report the performance when input encoding dimension is 1024 (L=31) for the sake of limited time.

| Setting | 1 km | 25 km | 200 km | 750 km | 2500 km |
|---|---|---|---|---|---|
| $w = 32, d = 6$ | 5.1 | 27.2 | 50.9 | 71.2 | 84.1 |
| $w = 128, d = 6$ | 4.7 | 27.0 | 50.2 | 70.8 | 84.3 |

