# OpenReview forum: "LocDiffusion: Identifying Locations on Earth by Diffusing in the Hilbert Space"
_ICLR.cc/2025/Conference — Submitted to ICLR 2025_

### Official Review · Reviewer_S6YX · 2024-10-31

**Soundness:** 3
**Presentation:** 4
**Contribution:** 3
**Rating:** 6
**Confidence:** 3

**Summary:**

"This paper proposes a novel spherical positional encoding-decoding scheme to address non-linearity issues, enabling effective optimization and training for geolocation tasks within a diffusion-based model. Additionally, it introduces a new conditional latent diffusion model, LocDiffusion, which achieves state-of-the-art performance in image geolocalization."

**Strengths:**

1. This paper introduces a noveldiffusion model to tackle the image geolocalization task, potentially benefiting the field and inspiring new directions.
2. The paper presents a new, learning-free encoder-decoder framework, mathematically demonstrating its robustness against sparsity issues in encoding space. This approach may also be beneficial for other remote sensing or geographical applications.
3. The paper provides solid mathematical proofs to support the properties it aims to achieve, particularly addressing issues of non-linearity and sparsity
4. The proposed approach achieves state-of-the-art results at the Region, Country, and Continent levels.
5. The paper is clear and well-written.

**Weaknesses:**

1. The ablation study lacks sufficient detail. For instance, it would be helpful to see an ablation on SHDD encoding and decoding—how much improvement does it provide over positional encoding or other encoder-decoder methods? Additionally, is the KL loss significantly more effective than L1 or L2 loss? Further clarification on the contributions of the other proposed modules would strengthen the analysis.

2. The model's performance at the street and city levels is relatively weak and does not achieve competitive results.

3. This method requires huge computational cost, limited to the upbound of L.

4. Need more details about computational cost, training time and inference time.

**Questions:**

1. In line 321, I noticed that $e_{lm}$ is used, but I couldn’t find a clear definition of it. Is $e_{lm}$ calculated from $F$? Could you clarify this or give the formula?

2. Is it reasonable to interpret the non-linear mapping between the locational embedding and position encoding space in Figure 1 as evidence of SHDD encoding’s superiority over traditional positional encoding? For my perspective, it is unconvincing, as the diffusion model introduces a non-linear mapping between $e$ and the image feature". Can you give more explanation about my confusion?

**Details Of Ethics Concerns:**

No concerns

---

> ### Author Response · Authors · 2024-11-28
> **Thank you very much for the comments and questions!**
>
> We try to address your concerns in the following
>
> **W1. Ablations**
>
> Thank you very much for giving such detailed suggestions on what ablations are missing. We have included all the three sets of ablation studies in Appendix A.6 and please kindly refer to our General Response 5 for a summary. Please kindly let us know if you find any part still missing in the analysis.
>
> **W2. performance at the street and city levels is relatively weak**
>
> We managed to improve the performance of LocDiffusion, especially on finer scales (please see our updated Table 1 in the main paper). Please kindly see our General Response 1. We have now added PIGEON to our baseline and properly cited it in the paper.
>
> **W3. limited to the upbound of L (low-pass filter)**
>
> Please kindly refer to our General Response 3. We developed some techniques to reduce the dimensionality of SHDD encoding. Also, for a general discussion of time and space complexity of our method, please kindly refer to our General Response 4.
>
> **W4. details about computational cost, training time and inference time**
>
> Please kindly refer to our General Response 4.
>
> **Q1. $e_{lm}$**
>
> Sorry for the confusion. The $e_{lm}$ is the SHDD encoding *generated* by the diffusion model; it is not calculated from $F$.
>
> **Q2. non-linear mapping between the locational embedding and position encoding space in Figure 1 as evidence of SHDD encoding’s superiority over traditional positional encoding?**
>
> The non-linear mapping here is the challenge for the alternative method based on location encoding. Since it is very hard to learn a neural net to accurately approximate such nonlinear mapping (see added Table 6 in A.6.1). One main benefit of  SHDD is to avoid such a non-linear mapping in the decoding step once the diffusion step is done -- the diffusion process itself is still highly nonlinear.

---

> > ### Comment · Reviewer_S6YX · 2024-12-02
> >
> > Thanks for your response, after reading more additional ablations, I will keep my score.

---

### Official Review · Reviewer_X6VD · 2024-11-03

**Soundness:** 1
**Presentation:** 2
**Contribution:** 2
**Rating:** 3
**Confidence:** 4

**Summary:**

This work proposes a diffusion model for image geolocalization (given an image, estimate where in the world the photograph was taken). Most work in this domain is based on either classification (partitioning the world into bins) or image retrieval. The main claim is that it overcomes limitations in spatial resolution caused by the partitioning scheme or the spatial distribution of gallery images. An encoding space is proposed that enables diffusion modeling on the sphere with the goal of reducing artifacts caused by modeling the sphere as a linear space.

**Strengths:**

- It's interesting to consider alternative formulations of long-standing problems.
- The proposed approach seems to work well for low-resolution localization.

**Weaknesses:**

- The work fails to deliver on its promise of eliminating the spatial resolution issues of previous approaches. Notably, Table 1 shows that it does not outperform approaches from 2016/2017 for low-error thresholds. The challenge is the computational complexity and memory consumption of the proposed approach. While promising, I'd need to see solid evidence that it improves relative to more straightforward approaches.

- L282 seems to be the crux of the problem with this approach, the assumption that the space is infinite. In practical implementations this is truncated, but it's unclear that this space is then the optimal space for the task. Existing approaches allow more data-driven specification of the space (for example, choosing spatial partitioning based on the distribution of the training dataset or implicitly based on the distribution of the gallery).

- L509 makes the claim that retrieval approaches depend heavily on the quality of the gallery, and seems to frame this as a negative. This is similar to saying, "the problem with the ImageNet dataset is that it requires so many images to represent the classes". Of course that's true; we always would love to use fewer images, but it's unclear how the proposed approach overcomes this issue. It still needs a large training dataset to fit the target function and these training images can serve as gallery images. There's clearly a negative to a large gallery, but in practice, it's easy to create indexes across billions of image embeddings using existing libraries. This scales well for practical uses.

- Some previous works have proposed strategies for overcoming the resolution limitations imposed by binning and a small gallery. This includes kernel density estimation in Revisiting IM2GPS in the Deep Learning Era by Nam Vo et al. A more complete assessment of previous attempts at addressing this issue should be included and an evaluation of simpler strategies should be included. Those missing baselines would likely be simpler to implement and might not suffer from the same memory challenges.

- Important related work is missing from CVPR 2024: "PIGEON: Predicting Image Geolocations" by Haas et al. This work proposes a hybrid of classification and retrieval which outperforms the results shown in Table 1 across the board. It notably does not generate poor results for low-error thresholds (which are the settings that most use cases require).

**Questions:**

I don't have any particular questions, but would be happy to hear discussion regarding the weaknesses identified above.

The writing style made the key contributions and approach difficult to follow. This is a matter of personal style, but here are a few notes / suggestions from my reading of the work:
  - There's quite a bit of mathematical development that seems to obscure the contribution. For example, I found myself skipping over Section 4.1 because it felt like it was being overly mathy without any strong motivation.
  - I'd suggest for a future submission that the authors consider bringing Figure 2 to the forefront and emphasizing the high-level implementation before delving into the detailed mathematics.
  - Figure 1 has the longest caption I've ever seen in a CV/AI/ML paper. Many of those details would have been better presented after the high-level algorithm was clear as opposed to so early in the paper.
  - I find the heavy use of footnotes to be a bad sign for the writing of a paper. It indicates to me that some concepts are being presented out of place.

---

> ### Author Response · Authors · 2024-11-28
> **W1:  fails to deliver on its promise of eliminating the spatial resolution issues of previous approaches.**
>
> Sorry for causing the confusion in our writing. Here by "arbitrary spatial resolution" we mean that our SHDD representation is a continuous function and can be evaluated at any arbitrary location, without dependence on any pre-defined gallery or bins. It is overloaded with the concept of "spatial resolution of SHDD decoding" where $L$ is the controlling factor. We sincerely apologize for the overload of concepts and we will replace the “arbitrary spatial resolution” term with “can be evaluated at arbitrary locations”. Please kindly see our General Response 2 where we present a detailed discussion.
>
> Besides, we have developed new techniques to increase $L$ and successfully improved the performance of LocDiffusion to outperform on finer scales. Besides, we find that hierarchically combing LocDiffusion with retrieval-based methods like GeoCLIP will yield even better results. Please kindly refer to the updated Table 1 in the main paper and our General Response 1 and General Response 3.

---

> ### Author Response · Authors · 2024-11-28
> **W2: in practical implementations L is truncated, but it's unclear that this space is then the optimal space for the task**
>
> The assumption of infinite coefficient space is a common and standard hypothesis used in many established fields, such as Fourier transformation. Where to truncate the vector is a trade-off between accuracy and computational efficiency.
> While we derive the SHDD encoding space from several desirable properties (density and less non-linearity), we do not claim that the SHDD space is optimal in all cases. The reason why we do not adopt a more data-driven approach is because we wish to
> (1) restrict the data-driven part to the diffusion model, so that the encoding/decoding process is learning-free, which is more stable and robust, and (2) make the method more spatially generalizable because the spatial distribution of training data may not align well with test data, as is seen in how GeoCLIP performs significantly worse on GWS15K dataset).

---

> ### Author Response · Authors · 2024-11-28
> **W3: still need a large "gallery" for training and inference**
>
> Your observation is correct that our approach still need a "gallery" for training and having a "gallery" also improves inference quality (e.g., when combining SHDD with GeoCLIP as in our General Response 1)
>
> However, our goal in this research is for the prediction accuracy not be constrained by the gallery -- i.e., requiring a gallery with examples co-locate with the test example. Please kindly see our General Response 2 where we present a detailed discussion.

---

> ### Author Response · Authors · 2024-11-28
> **W4. & W5. comparing to PIGEON and simpler strategies**
>
> Thank you very much for pointing us to the previous works we have unfortunately overlooked. We have added PIGEON to our baselines and properly cited it in our updated paper. The hybrid of classification and retrieval is very inspiring to us: we tried the hybrid of generation and retrieval (i.e., use LocDiffusion to generate a vague guess of image distributions, just like the classification step in PIGEON, and use GeoCLIP to retrieve in this guessed area) and  saw performance improvement over both methods.
> This indicate a very promising direction to combine the strong spatial generalizability of generative models and the strong fine-grained localization of retrieval-based models.
>
> As for the simpler strategies, while we were unable fully evaluate them due to the limited rebuttal time, we are happy to include them as baselines in the camera-ready version.

---

> ### Author Response · Authors · 2024-11-28
> **Writing suggestions**
>
> Thank you for the suggestions on the paper writing. We are happy to make the following changes to the paper structure in the camera-ready version:
>
> (1) The motivation of Section 4.1 is based on the two concepts discussed in the introduction: sparsity and non-linearity. We try to use Section 4.1 to formally define what sparsity and non-linearity are and use math to derive what kind of encoding/decoding we need to overcome sparsity and non-linearity. We can write out this motivation more explicitly and link this section more obviously with both the introduction and the method sections.
>
> (2) We will move Figure 2 to introduction.
>
> (3) We will split Figure 1 into several subfigures and move the captions to the main contents.
>
> (4) We will try to reduce the use of footnotes, e.g. move it to Appendix.

---

### Official Review · Reviewer_u6sy · 2024-11-03

**Soundness:** 3
**Presentation:** 3
**Contribution:** 3
**Rating:** 8
**Confidence:** 4

**Summary:**

This paper tries to address the task of image geolocalization that estimates the geographic location where an image was taken. This paper introduces LocDiffusion, a novel generative approach to image geolocalization that leverages diffusion models to predict locations on Earth from images. The key innovation is a new Spherical Harmonics Dirac Delta (SHDD) encoding-decoding framework that enables diffusion in a Hilbert space while preserving spherical geometry. The authors also propose a Conditional Siren-UNet architecture for learning the conditional backward diffusion process.

**Strengths:**

1.	Creative solution to the manifold projection problem in diffusion models.
2.	Novel SHDD encoding framework that handles spherical geometry.
3.	While the concept of KL-divergence itself is not new, the authors' specific application of it to spherical location generation through SHDD encoding and decoding appears to novel.
4.	The proposed Spherical Harmonics Dirac Delta (SHDD) encoding-decoding framework tackles the challenges of traditional image geo-localization methods by enabling the use of diffusion models for generating locations on the spherical surface of the Earth.

**Weaknesses:**

1.	Quadratic scaling of SHDD encoding dimension may limit practical resolution.
2.	High space complexity and long training process.
3.	No ablation study.

**Questions:**

1.	The main purpose of image-based geo-localization is to predict the location of an image. Unlike image retrieval-based localization, can this method predicate the coordinates?
2.	Unlike using a large model for geo-localization, large model in general (even CLIP), has powerful prior knowledge, and therefore can discern the approximate location when the landmark is given. However, it just only predicts approximate locations, not coordinates. To build such a geographic knowledge base, a huge number of images are needed.
3.	I noticed some tricks in preventing overfitting phenomena, such as a very high dropout rate. Please analyze the overfitting of the proposed method.
4.	the advantages of the SHDD representation are primarily presented at a theoretical level, with limited experimental evidence to fully support the claims, without more experiments or analysis. By the way, there are only three tables in the entire paper, and one of them is still or just an experimental setup. There is no experimental-level evidence of the advantages of model and only a minor improvement in the main experiment.
5.	As shown in Table 1, in YFCC-26k, GeoDecoder has better performance than GeoCLIP in City 25km. I would recommend labeling the strong petitory and marking the improvement over them. Why is the performance so defective in 1km and 25km? The increase in generalization ability is built at the expense of model accuracy, which is hard to accept. I understand the need for generalization ability for this task, but I can only assume that for the previous models are flawed in solving the fitting problem. If the main purpose of this paper is in addressing the generalization ability, it would be more appropriate to analyze this aspect or collect an open-world test set to validate the model.

---

> ### Author Response · Authors · 2024-11-28
> **Thank you very much for the comments and questions!**
>
> We try to address your concerns in the following
>
> **W1. Quadratic scaling of SHDD encoding dimension**
>
> Please kindly refer to our General Response 3 where we developed techniques to reduce the dimensionality of SHDD encoding.
>
> **W2. High space complexity and long training process.**
>
> Please kindly refer to our General Response 4 for detailed computational efficiency analysis. We believe that both space and training complexities can be further improved in the future.
>
> **W3. Ablations**
>
> We have added more ablation experiments to help understand the behavior of the proposed model.
> Please kindly refer to our General Response 5 for detailed ablations of different components.

---

> ### Author Response · Authors · 2024-11-28
> **Responses to questions**
>
> **Q1: Unlike image retrieval-based localization, can this method predicate the coordinates?**
>
> Yes. The original motivation is to get rid of discrete, pre-defined galleries and directly generate real-valued location coordinates.
>
> **Q2: a huge number of images are needed to build the model?**
>
> The image dataset we train LocDiffusion on is 4 million large in size, which is a "small" dataset compared to the real large models. The reason we can do this is because we use the pretrained CLIP embeddings as image input. In other words, our model is a "task-specific finetuning" on top of large models, so we need much less data (since a lot of prior knowledge is inherited from CLIP).
>
> **Q3.  analyze the overfitting**
>
> The overfitting phenomena, from our observation, seem to originate from SHDD encoding. Please see Figure 4(b) in Appendix A.3. While for an $L$ degree SHDD encoding there are $(L+1)^2$ dimensions, many dimensions have very small coefficient values and contribute little to the shape of the function. The diffusion model may overfit on these dimensions and pay less attention to real important dimensions. This is just a hypothesis but we are happy to explore more into this issue.
>
> **Q4. minor improvement in the main experiment.**
>
> We managed to improve the performance of LocDiffusion, especially on finer scales (please see our updated Table 1 in the main paper). Please kindly see our General Response 1. We have now added PIGEON to our baseline and properly cited it in the paper. We also added a section of ablation experiments in Appendix 6. Please kindly refer to our General Response 5. We tested how different choices of location encoder/decoder, different choices of loss functions and different architecture settings may affect the performance. Please let us know if you find any part missing and we are more than happy to add it to the paper.
>
> **Q5. why performance so defective in 1km and 25km?**
>
> The defective performance on fine-grained scales in the previous experiment results originates from using inadequately large $L$, since $L$ controls the spatial resolution of SHDD decoding, i.e., scales smaller than this spatial resolution can not be accurately decoded. In the updated paper, we managed to increase $L$ and improve the 1km and 25km scale performance. Further, we provide a detailed analysis of how and why retrieval-based methods like GeoCLIP suffer from low spatial generalizability in Appendix A.4. Please kindly refer to our General Response 1 and General Response 5 for more detailed discussions.
>
> Besides, we find that hierarchically combine LocDiffusion with retrieval-based methods can exploit their respective advantages and make more performance improvement. Briefly speaking, we use LocDiffusion to generate a set of coarse guesses of the image locations, and use GeoCLIP to retrieve the location prediction from the neighorhoods of the coarse guesses. This joins the spatial generalizability of LocDiffusion (not needing a global level gallery) and the fine-grained scale accuracy of GeoCLIP. Please kindly see our updated Table 1 in the main paper and our General Response 1, General Response 2 for more details.

---

### Official Review · Reviewer_Be4r · 2024-11-04

**Soundness:** 3
**Presentation:** 3
**Contribution:** 2
**Rating:** 3
**Confidence:** 4

**Summary:**

1. Authors proposed a spherical positional encoding-decoding framework, which encodes points on a spherical surface into a Hilbert space of Spherical Harmonics coefficients.

2. Authors proposed a UNet architecture to learn conditional diffusion by minimizing a latent KL-divergence loss and train a LocDiffusion model that addresses the image geolocalization task via generation.

**Strengths:**

S1: This article is well-written and easy to understand.

S2: This paper is the first work (to my knowledge) to use a diffusion model for image geolocalization.

(I also suggest that the authors change the word "generative" in the relevant claims to "diffusion ".)

**Weaknesses:**

W1: This paper has a lot of descriptions of methods, but the experiments are insufficient to support the discussion of the methods. The authors need more ablation experiments. (I can revise the scores based on feedback from the authors.)

W2: Authors have a strong theoretical basis for the design of the method, but unfortunately, the experimental results are not significantly improved, whether it is GeoCLIP compared in the paper or PIGEON, which is not cited.

@InProceedings{Haas_2024_CVPR,
    author    = {Haas, Lukas and Skreta, Michal and Alberti, Silas and Finn, Chelsea},
    title     = {PIGEON: Predicting Image Geolocations},
    booktitle = {Proceedings of the IEEE/CVF Conference on Computer Vision and Pattern Recognition (CVPR)},
    month     = {June},
    year      = {2024},
    pages     = {12893-12902}
}

**Questions:**

Q1: Can you perform qualitative or quantitative experiments comparing the proposed positional encoding with other methods in the supplementary material?

Q2: Can you provide some ablation experiments?

---

> ### Author Response · Authors · 2024-11-28
> **Thank you very much for the comments and questions!**
>
> We try to address your concerns in the following
>
> **S2: change the word [generative] in the relevant claims to [diffusion]**
>
> Thank you for this suggestion! We will change the wording accordingly.
>
> **W1. more ablation experiments**
>
> We added a section of ablation experiments in Appendix 6. Please kindly refer to our General Response 5. We tested how different choices of location encoder/decoder, different choices of loss functions and different architecture settings may affect the performance. Please let us know if you find any part missing and we are more than happy to add it to the paper.
>
> **W2. experimental results are not significantly improved**
>
> By increasing L in a memory efficient way and combining with GeoCLIP retrieval we managed to improve the performance of LocDiffusion, especially on finer scales (please see our updated Table 1 in the main paper). Please kindly see our General Response 1. We have now added PIGEON to our baseline and properly cited it in the paper.
>
> **Q1./Q2. ablations**
>
> Please kindly refer to our General Response 5 and the Appendix A.6 in the updated paper.

---

> > ### Comment · Reviewer_Be4r · 2024-11-28
> >
> > Thank you for your feedback. I have updated my score.

---

### Official Review · Reviewer_GNES · 2024-11-05

**Soundness:** 2
**Presentation:** 3
**Contribution:** 3
**Rating:** 6
**Confidence:** 3

**Summary:**

This paper introduces a latent diffusion model called LocDiffusion for image geolocalization, aiming to predict precise geolocation on Earth from images with arbitrary resolutions. It proposes a novel spherical position encoding method, Spherical Harmonics Dirac Delta (SHDD) Representation, which encodes geocoordinates on Earth's spherical surface into a Hilbert space of Spherical Harmonics coefficients. The model then decodes these geolocations via a mode-seeking approach. A SirenNet-based architecture is used to learn the conditional backward process in the latent SHDD space by minimizing a latent KL-divergence loss. Experiments are conducted on the Im2GPS3k and YFCC-26k datasets, with comparisons to state-of-the-art methods like GeoCLIP.

**Strengths:**

1. This work is the first generative model addressing the image geolocalization problem, offering a fresh perspective to the field.
2. The design of the Spherical Harmonics Dirac Delta (SHDD) Representation is innovative and tailored to the geocoordinate system, which is both novel and meaningful.
3. The paper is well-written and generally easy to follow.

**Weaknesses:**

1. The performance gains over state-of-the-art methods, particularly GeoCLIP, are not well demonstrated. In Table 1, the proposed method achieves the best results in 5 out of 10 categories across two benchmarks, but the improvements are marginal, with differences of less than 0.5. Additionally, the method significantly underperforms in key categories like Street, City, and Region, raising concerns about its robustness across different geolocalization tasks.
2. The paper lacks a detailed analysis of computational efficiency, particularly in terms of training and inference speed, compared to existing methods. Since this is the first application of a diffusion model for geolocalization, it is important to understand how it compares to more established models, such as CLIP-based approaches, in terms of scalability and practicality for real-world applications. This could also helping assessing its potential impact and usability.

**Questions:**

1. In Lines 92-93: The paper states, "Performing diffusion on the XYZ coordinates will likely lead to a point that is not on the spherical surface." Can the authors please provide further explanation or supporting evidence for this claim?
2. In Table 1, the method appears to struggle when the distance is below 1/3 or 1/10 of the spatial threshold in the SHDD representations. Could the authors explain why this occurs, especially given the claim that the model works for images with arbitrary resolutions? Does this limitation suggest the model is better suited for coarse-level geolocalization rather than fine-grained predictions?
3. Increasing the parameter $L$ seems to have a significant impact on performance. Could the authors provide more details on the computational resources required as $L$ increases? Understanding the trade-offs between computational cost and performance would offer useful insights into the model's scalability.

**Details Of Ethics Concerns:**

Geolocalization models, particularly those capable of predicting precise locations from images, can raise significant privacy concerns if misused. Such models could unintentionally expose the locations of individuals or sensitive areas without consent, potentially leading to privacy violations. This paper does not address these ethical implications, which is an important consideration that should be discussed and included to ensure responsible use of the technology.

---

> ### Author Response · Authors · 2024-11-28
> **Thank you very much for the comments and questions!**
>
> We try to address your concerns in the following
>
> **W1. Marginal performance gain.**
>
> The marginal performance gain presented in the paper is because we used a relatively small degree of Legendre polynomials $L=31$ since our computational resources were limited. Analysis shows the spatial resolution at this degree is around 320km and it is very difficult to achieve good performance on fine-grained scales. We developed a low-pass filtering technique to reduce the quadratic dimensionality of SHDD and we successfully used larger $L=47$ and saw significant further performance improvements. We also found that hierarchically combining LocDiffusion and GeoCLIP yields better performance that performs better than both. Please kindly refer to the updated Table 1 in the main paper and our General Response 1 for detailed discussions.
>
> The main benefit of the diffusion approach is to avoid the assumption that training and testing points have significant co-locations. Without this assumption the retrieval based approaches perform extremely poorly as demonstrated in Table 2 and Appendix A.4. Please see detailed relay in our General Response 2.
>
> **W2. Lack of complexity analysis**
>
> Please kindly refer to our General Response 4 for a discussion on the complexity of LocDiffusion.
>
> **Q1. Why performing diffusion on the XYZ coordinates will likely lead to a point that is not on the spherical surface.**
>
> Considering representing a point on the sphere using its x-y-z coordinates. If we add a Gaussian noise to this triplet, it is almost surely that the resulted point is no longer on sphere. For example, if the origin (0, 0, 0) is at the north pole, and you add a random noise (0, 0, 0.1) to it. Obviously (0, 0, 0.1) is not on the sphere: it is 0.1 units above the sphere along the z-axis. Abstractly speaking, it is because a sphere is a zero-measure manifold embedded in the 3-D Euclidean space.
>
> **Q2. the trade-off between computational stability and spatial resolutions of SHDD decoding.**
>
> Please refer to our General Response 1 for detailed discussions. In general, we find that combing LocDiffusion (coarse predictions) and retrieval-based methods (extremely fine predictions) may yield better results than using either alone.
>
> **Q3. Increasing the parameter L**
>
> Please kindly refer to our General Response 1 and General Response 3 for detailed discussions. In Appendix A.3 we carefully studied how $L$ influences the performance and we developed a low-pass filtering technique to improve model performance at lower expenses.
>
> **Ethic Concerns**
>
> Thank you for pointing out this issue. We will update our Ethics Statement to discuss risks.
>
> Image geolocalization comes with both potential benefits and risks of misuse.  On the one hand, accurate geo-tagging of images opens up possibilities for various beneficial applications such as autonomous driving, navigation, geography education, open-source intelligence, and visual investigations in journalism. On the other hand, however, geo-tagging also comes with risks, such as military uses and privacy risks. Today’s image geolocalization technologies are unable to make street-level predictions reliably, which reduces their applicability to the above risks. Nevertheless researchers should be aware that such technologies are becoming increasingly precise.

---

### Author Response · Authors · 2024-11-28
**General Response 1. Improving the performance of LocDiffusion**

**General Response 1. Improving the performance of LocDiffusion**

As all the reviewers have noticed and brought up in the comments, the performance of LocDiffusion compared with existing retrieval-based methods such as GeoCLIP and PIGEON was not ideal, especially on low-error scales such as 1km and 25km. We were also aware of this and managed to improve it to be on pare with or even outperform SOTA methods. Please see our updated Table 1 in the revised PDF.

The major factor that constrained the performance of LocDiffusion in the previous experiment results is the Legendre polynomial degree $L$. As we have discussed in Section 4.2 and Appendix A.3, the spatial resolution of SHDD decoding, i.e., the variance of SHDD decoding introduced by using a truncated spherical harmonics vector, is dependent on $L$. See Figure 3 in Appendix A.3 for an illustration. In our previous experiment results, while we were aware that larger $L$ brings better performance (See Figure 4(a) in  Appendix A.3), we only used $L = 31$ (i.e., SHDD encoding dimension = 1024) due to limited computational resources. The spatial resolution is around 320km. It is therefore very difficult to achieve high performance on fine-grained scales such as 1km and 25km because the intrinsic variance of the decoded locations is much larger.

To overcome this, we developed a dimension reduction technique to effectively use larger $L$. Briefly speaking, we find that not all $(L+1)^2$ dimensions are equally important in the computation of KL-divergence and in the decoding. See Figure 4(b) in Appendix A.3 for an illustration. Just like in signal processing, the Fourier features that have very small magnitudes can be discarded by a low-pass filter without significantly changing the shape of the signal; we applied the similar idea to the SHDD encoding. For example $L=47$ correspond to a 2403-dimensional vector, and we only kept the 1024 dimensions whose spherical Harmonics terms have top absolute coefficient values. In this way we were able to increase the spatial resolution of LocDiffusion and achieve further performance improvements.

However, further increasing $L$ leads to numerical instabilities because the coefficient values get too large and overflow when we calculate their exponentials for probability computation. As shown in Figure 4(b) in Appendix A.3, 2500 is the rough upper-bound of $L$ where we can stably compute the KL-divergence loss and decode an SHDD encoding. Therefore we stopped at 2403 dimensions. Whether we can overcome this hurdle and use even higher spatial resolutions is an interesting future research problem. However, the spatial resolution of $L=47$ is around 200 km, which is still significantly larger than 1km and 25km. Thus, while the performance of LocDiffusion on these two scales is close to GeoCLIP but can not beat it yet.

Given the success of PIGEON (which is a hierarchical geolocalization framework), we explore combining LocDiffusion and GeoCLIP: since LocDiffusion is a generative model and outperforms GeoCLIP on coarser scales (> 200 km) while GeoCLIP outperforms on finer scales (<25km), we can use it as the first stage of geolocalization hierarchically. First, we sample multiple times (e.g., 16) and get a rough distribution of candidate locations, i.e. they indicate where the true answer is highly likely to reside. Then, we restrict the retrieval of GeoCLIP to the neighborhoods (200 km radius) of these candidate locations. We find that this simple combination yields better results than both GeoCLIP and LocDiffusion used alone. This approach is to  recommendation systems' retrieve and rerank approach. This combination points to interesting future work, too, and we wish to demonstrate that **our generative approach is not meant to beat and replace retrieval-based models; instead, they have their respective advantages and the best strategy is to fuse them into one system**.

---

### Author Response · Authors · 2024-11-28
**General Response 2. Clarification on the contribution of the paper**

**General Response 2. Clarification on the contribution of the paper**

Our goal in this research is not to beat and replace retrieval/classification based models; instead, we wish to provide a **more spatially generalizable alternative** and align it with previous works that try to overcome the resolution limitations of bins and galleries. More importantly, apart from the specific application to geolocalization, the design of SHDD encoding/decoding techniques itself may open up research opportunities in other applications where decoding locations is required (as far as we know, it is the first deterministic, learning-free spherical location decoding method).

Here we demonstrate that 1) how existing approaches rely on co-locations between training and testing data and why it may be problematic, 2) how our approach does not suffer from this problem (as demonstrated in Table 2 and Appendix A.4) and 3) by hierarchically using our model as the first stage of retrieval we can improve SOTA approaches (e.g., GeoCLIP) at all scales.

First, we'd like to illustrate that the existing retrieval process used by GeoCLIP (as in Table 1) implicitly relies on a strong assumption that many test images co-locate with the retrieval gallery. To demonstrate, we take the MP16 gallery used by GeoCLIP and compare the locations in the gallery with the locations in the Im2GPS3K test set. We measure how closely test locations align with the gallery locations by counting the number of gallery locations that are within 1km/25km from a given test location. There are 63.5% test locations that can find at least one location in the MP16 gallery that is less than 1 km away, and 95.2% less than 25 km away. **This means the “correct answer” is almost always present in the gallery – this is a very strong spatial inductive bias**. If we do not have access to such a gallery, for example if we use a set of evenly spaced locations on Earth as the gallery, the two numbers drop to 0.1% 1km away and 38.9% 25 km away, respectively. A detailed analysis can be found in Appendix A.4. This can be very problematic since **in real-world settings, such a well-aligned gallery is hard to obtain**. GWS15K is an actual example. In the GeoCLIP paper, the low-error resolution performance on GWS15K is significantly worse (0.6 on 1km, 3.1 on 25km) than on Im2GPS3K (14.11 on 1km, 34.47 on 25km). However, both GeoCLIP and PIGEON did not publicize this dataset, and to construct it using Google APIs takes longer time beyond the rebuttal period. We are happy to include this result in the camera-ready version if this paper gets accepted.

Second, we’d like to explain why our LocDiffusion model does not have this issue. SHDD encoding is an implicit representation of a **continuous function**. One can **exactly** evaluate the function values at any point on the sphere without the need of any gallery or spatial bins; the only factor that restricts the accuracy of the evaluation is the computational setting (i.e., using how many digits of floats). Instead, retrieval-based methods can only be applied to locations that exist in the gallery. In this sense, our claim in the paper that our method can achieve **arbitrary spatial resolution** can be confusing comparing to the **spatial resolution of SHDD decoding** we discussed in General Response 1. We sincerely apologize for the overload of concepts and we will replace the “arbitrary spatial resolution” term with “can be evaluated at arbitrary locations”. The benefit of being independent of a pre-defined gallery is stronger spatial generalization. Table 2 in the main paper and Figure 5 in Appendix A.4 clearly demonstrate that LocDiffusion performs stably well no matter whether the SHDD decoding is evaluated on MP16 gallery points or on evenly distributed grid points.

Finally, as we have also discussed in General Response. 1, **the best strategy may be a hierarchical combination of real-valued diffusion and discrete retrieval/classification (like that of PIGEON): ** use diffusion in coarser scales (>200km) to accurately narrow down the region the image may locate and for further fine scales (< 25km) use a locally trained retrieval model. This will help avoid retrieving from an extremely large gallery that might introduce noise retrieval results because of the needs to be both global and spatially fine-grained. The performance gain of such a combination is presented in the updated Table 1.

---

### Author Response · Authors · 2024-11-28
**General Response 3. Addressing the quadratic complexity of SHDD encoding**

**General Response 3. Addressing the quadratic complexity of SHDD encoding**

The Legendre polynomial degree $L$ controls the sizes of spherical harmonics wavelengths and has a significant impact on the resolution (See Figure 4(a) in Appendix A.3). Several reviewers are rightfully concerned that the dimension of SHDD encoding scales quadratically. We developed a low-pass filtering technique to effectively reduce the dimensionality.

Briefly speaking, we find that not all $(L+1)^2$ dimensions are equally important in the computation of KL-divergence and in the decoding. See Figure 4(b) in Appendix A.3 for an illustration. Just like in signal processing, the Fourier features that have very small magnitudes can be discarded by a low-pass filter without significantly changing the shape of the signal; we applied the similar thing to the SHDD encoding. We used $L=47$, got a 2403-dimensional vector, and only kept the 1024 dimensions whose spherical Harmonics terms have top absolute coefficient values. In this way we were able to increase the spatial resolution of LocDiffusion and saw significant performance improvements especially for scale 1km and 25km.

Another possible approach is analogous to nested dropout. For each pass we only evaluate a fixed-size random subset of all the dimensions in hope that as training epochs increase, all dimensions are visited and learned.

---

### Author Response · Authors · 2024-11-28
**General Response 4. Computational complexity**

**General Response 4. Computational complexity**

The computational complexity of LocDiffusion, like most diffusion models, lies mainly in the diffusion process. **The SHDD encoding/decoding is deterministic and learning-free, which is very fast**. In fact, if the training data are fixed, we only need to construct a look-up table. Therefore, the time complexity of LocDiffusion is at the same level as the specific diffusion model we employ (for example, in this paper we used the classic DDPM model without sampling acceleration). We potentially can apply more efficient diffusion approaches such as stable diffusion or consistency diffusion.

The space complexity depends on the input and output dimensions of the diffusion model, i.e., the dimensions of SHDD encodings. As we have discussed in General Response 1 and General Response 3, it is both possible to reduce the dimensionality of SHDD encodings by low-pass filtering, and recommended not to use very high dimensional SHDD encodings for the sake of computational stability.

For details on the training and inference time/space costs, please see Appendix A.5. The inference time of LocDiffusion is about 0.05s per query on a single NVIDIA RTX5500 24G GPU, while GeoCLIP about 0.02s. Therefore in practice, the computational complexity of LocDiffusion should not be a huge concern. If we need to scale up, we can apply the well-established sampling acceleration techniques developed for diffusion models.

---

### Author Response · Authors · 2024-11-28
**General Response 5. Ablation studies**

**General Response 5. Ablation studies**
In Appendix A.6 we have added more ablation experiments to help understand the behavior of the proposed model.
There are three sets of ablation experiments:

**(1) Ablation on replacing SHDD with other location encoding/decoding methods**

Theoretically we argued in Section 4.1 that existing location encoding methods suffer from sparsity and non-linearity so that they are not suitable for location decoding. This is supported by ablation studies. Please see Table 6 and Table 7 in Appendix A.6. To summarize, we replace SHDD encoder/decoder with two commonly used location encoders (rbf and Sphere2Vec) and train a neural network to decode the diffusion output back to locations. After the replacement, the geolocalization performance is reduced by half on coarser scales, and on finer scales (1km and 25km) the accuracy drops to 0. We further demonstrate that such performance drop comes partially from the non-linearity of these location encodings (another reason is that the neural decoder is not 100% accurate). Adding a small Gaussian noise (variance = 0.01) to a non-SHDD location encoding, the spatial drift of the decoded location can be as large as 102.4km (rbf) and 89.1km (Sphere2Vec), while for SHDD encoding this drift is 5.3km. That means, in the case that the diffusion generated output is not noiseless (which is almost certain), SHDD suffers less.

**(2) Ablation on replacing SHDD KL-divergence loss with other losses**

We try to train LocDiffusion with alternative losses, such as L1/L2 and cosine distance loss. Please see Table 8 in Appendix A.6. The findings are very interesting: L1/L2 losses are much worse than SHDD KL-divergence, whereas cosine distance loss yields fairly good results on coarser scales. Our hypothesis is that the cosine distance loss is mathematically similar to the SHDD KL-divergence loss (SHDD KL-divergence is the sum of exponential element-wise multiplications, while cosine similarity is the sum of raw element-wise multiplications; the difference is that exponential terms are non-negative and the probability mass is more concentrated, i.e., it should be better than cosine losses on fine-grained scales, but not very different on coarser scales). This may indicate that we can use cosine loss as a computationally friendly alternative as long as we do not insist on high performance on low-error scales. This is also an interesting future work.

**(3) Ablation on CS-UNet modules**

We have explored many alternatives of the CS-UNet architecture, but unfortunately we did not record them for ablation studies. Due to the limited time given for rebuttal and the long training time and various combinations of alternative modules, we have not been able to conduct a very thorough ablation study on this part. But according to our experience, one of the most important factors is the bottleneck dimension of the CS-UNet: shrinking the bottleneck width w of the CS-UNet seems to help alleviate model overfitting (we can adopt a lower dropout rate) and slightly boost performance, but it makes the model more difficult to train. We managed to do this experiment and reported the results in Table 9 in Appendix A.6.

---

### Meta-Review · Area_Chair_AABd · 2024-12-20

**Metareview:**

The submission proposes LocDiffusion, a novel latent diffusion model for image geolocalization, leveraging a spherical positional encoding-decoding scheme (SHDD) to operate in a Hilbert space while preserving Earth's spherical geometry. While the paper presents innovative ideas and introduces a new generative framework for geolocalization, its contributions are undermined by limited experimental improvements, incomplete comparisons to related work, and unresolved concerns raised by reviewers during the discussion period.

Reviewers GNES, Be4r, and X6VD expressed concerns about whether the proposed approach offers tangible benefits over existing methods in terms of accuracy and practical usability. Missing or insufficiently analyzed baselines include well-established methods like PIGEON (CVPR 2024) and other retrieval-based geolocalization approaches. Reviewer X6VD highlighted these omissions as critical weaknesses, questioning the significance of the proposed contributions. The authors’ response addressed some writing concerns but failed to adequately clarify key methodological aspects, as noted by Reviewer S6YX.

**Additional Comments On Reviewer Discussion:**

Several reviewers (GNES, Be4r, X6VD) retained low scores after the discussion phase, citing unresolved concerns about the paper’s novelty, experimental results, and comparisons to related work. Reviewer Be4r raised concerns about the authors’ deliberate omission of key baselines, which could undermine the integrity of the work.

---

### Decision · Program_Chairs · 2025-01-22

Reject